# EMMA: End-to-End Multimodal Model for Autonomous Driving

**Jyh-Jing Hwang**\* †, **Runsheng Xu**\*, **Hubert Lin**‡, **Wei-Chih Hung**‡, **Jingwei Ji, Kristy Choi**
**Di Huang, Tong He, Paul Covington, Benjamin Sapp, Yin Zhou, James Guo**
**Dragomir Anguelov, Mingxing Tan**†

*Waymo LLC*

**Reviewed on OpenReview:** *https://openreview.net/forum?id=kH3t5lmOU8*

## Abstract

We introduce EMMA, an End-to-end Multimodal Model for Autonomous driving. Built upon a multi-modal large language model foundation like Gemini, EMMA directly maps raw camera sensor data into various driving-specific outputs, including planner trajectories, perception objects, and road graph elements. EMMA maximizes the utility of world knowledge from the pre-trained large language models, by representing all non-sensor inputs (e.g. navigation instructions and ego vehicle status) and outputs (e.g. trajectories and 3D locations) as natural language text. This approach allows EMMA to jointly process various driving tasks in a unified language space, and generate the outputs for each task using task-specific prompts. Empirically, we demonstrate EMMA's effectiveness by achieving state-of-the-art performance in motion planning on nuScenes as well as competitive results on the Waymo Open Motion Dataset (WOMD). EMMA also yields competitive results for camera-primary 3D object detection on the Waymo Open Dataset (WOD). We show that co-training EMMA with planner trajectories, object detection, and road graph tasks yields improvements across all three domains, highlighting EMMA's potential as a generalist model for autonomous driving applications. We hope that our results will inspire research to further evolve the state of the art in autonomous driving model architectures.

## 1 Introduction

Autonomous driving technology has made significant progress in recent years. To make autonomous vehicles a ubiquitous form of transportation, they must navigate increasingly complex real-world scenarios that require understanding rich scene context as well as sophisticated reasoning and decision-making.

Historically, autonomous driving systems employed a modular approach, consisting of specialized components for perception (Yurtsever et al., 2020; Li et al., 2022b; Lang et al., 2019; Sun et al., 2022; Hwang et al., 2022), mapping (Li et al., 2022a; Tancik et al., 2022), prediction (Nayakanti et al., 2023; Shi et al., 2024), and planning (Teng et al., 2023; Lioutas et al., 2022). While this design lends itself to easier debugging and optimization of individual modules, it poses scalability challenges due to the limited inter-module communication. In particular, the expert-designed interfaces between modules, such as the perception and behavior modules, may struggle to adapt to novel environments because they are often pre-defined based on targeted scenarios (Bansal et al., 2019; Jiang et al., 2023; Nayakanti et al., 2023; Seff et al., 2023). End-to-end autonomous driving systems (Hu et al., 2023; Zhai et al., 2023; Li et al., 2024) have recently emerged as a potential solution, directly learning to generate driving actions from sensor data. This approach eliminates the need for symbolic interfaces between modules and allows for joint optimization of driving objectives from

---

\*Equal contributions; ‡ Equal contributions.
†Contact emails: Mingxing Tan *<tanmingxing@waymo.com>*, Jyh-Jing Hwang *<jyhh@waymo.com>*.

raw sensor inputs. However, these systems are often specialized for specific driving tasks and trained on limited datasets, hindering their ability to generalize to rare or novel scenarios.

Multimodal Large Language Models (MLLMs) (Gemini Team Google, 2023; Achiam et al., 2023) offer a promising new paradigm for AI in autonomous driving that may help to address such challenges. This is because MLLMs, as generalist foundation models, excel in two key areas: (1) they are trained on vast, internet-scale datasets that provide rich "world knowledge" beyond what is contained in common driving logs, and (2) they demonstrate superior reasoning capabilities through techniques such as chain-of-thought reasoning (Wei et al., 2022; Zhang et al., 2023b) that are not available in specialized driving systems. While recent efforts (Chen et al., 2024b; Tian et al., 2024) have explored integrating and augmenting the capabilities of existing driving systems with MLLMs, we propose to develop an autonomous driving system in which the MLLM is a first class citizen.

We introduce the End-to-End Multimodal Model for Autonomous Driving (EMMA), built on top of a multimodal large language model, such as Gemini (Gemini Team Google, 2023) or PaLI (Chen et al., 2024d) without additional specialized components. Figure 1 shows the overview of the EMMA framework. EMMA accepts camera images and plain text for other non-vision inputs such as high-level driving commands and historical context. By recasting driving tasks as visual question answering (VQA) problems, EMMA leverages Gemini's pre-trained capabilities and extensive world knowledge. After EMMA is fine-tuned with driving logs from all tasks using task-specific prompts (see Figure 2 for more examples), it generates various driving outputs such as future trajectories for motion planning, perception objects, road graph elements, and scene semantics. Our experiments showcase EMMA's strong performance on several planning and perception benchmarks despite this simple design. Additionally, we find that EMMA can produce interpretable, human-readable outputs for many perception tasks such as road graph estimation, and is able to function as a generalist model that is both scalable and robust for autonomous driving. Notably, as used here and throughout the paper, the *EMMA generalist model* refers to a machine learning model that has been trained and fine-tuned on a large volume of driving data to perform a wide range of driving tasks in the autonomous driving domain.

We summarize our key findings below:

1. EMMA exhibits strong performance in **end-to-end motion planning**, achieving state-of-the-art performance on public benchmarks nuScenes (Caesar et al., 2020) and competitive results on the Waymo Open Motion Dataset (WOMD) (Chen et al., 2024a). We also show that we can further improve motion planning quality with more internal training data and chain-of-thought reasoning.

2. EMMA demonstrates competitive results for various perception tasks including **3D object detection, road graph estimation, and scene understanding**. On the camera-primary Waymo Open Dataset (WOD) (Hung et al., 2024), EMMA achieves better precision and recall for 3D object detection than state-of-the-art methods.

3. We demonstrate that EMMA can function as a **generalist model in the autonomous driving domain**, which jointly generates the outputs for multiple driving related tasks. In particular, EMMA matches or even surpasses the performance of individually trained models when it is co-trained with motion planning, object detection, and road graph tasks.

4. Finally, we show EMMA's capacity to reason and make decisions in **complex, long-tail driving scenarios**.

In the remainder of this paper, Section 2 describes the detailed method of EMMA for end-to-end motion planning and generalist tasks in autonomous driving. In Section 3, we present experimental results of EMMA on public and internal datasets. Finally, we discuss related works in Sections 4.

Despite these promising results, EMMA is not without its limitations. We discuss the limitations in-depth in the Appendix Section A.5. In particular, it faces challenges for real-world deployment due to: (1) limitations in 3D spatial reasoning due to its inability to fuse camera inputs with LiDAR or radar, (2) the need for realistic and computationally expensive sensor simulation to power its closed-loop evaluation, and (3) the increased computational requirements relative to conventional models. We plan to better understand and address such challenges in future work.

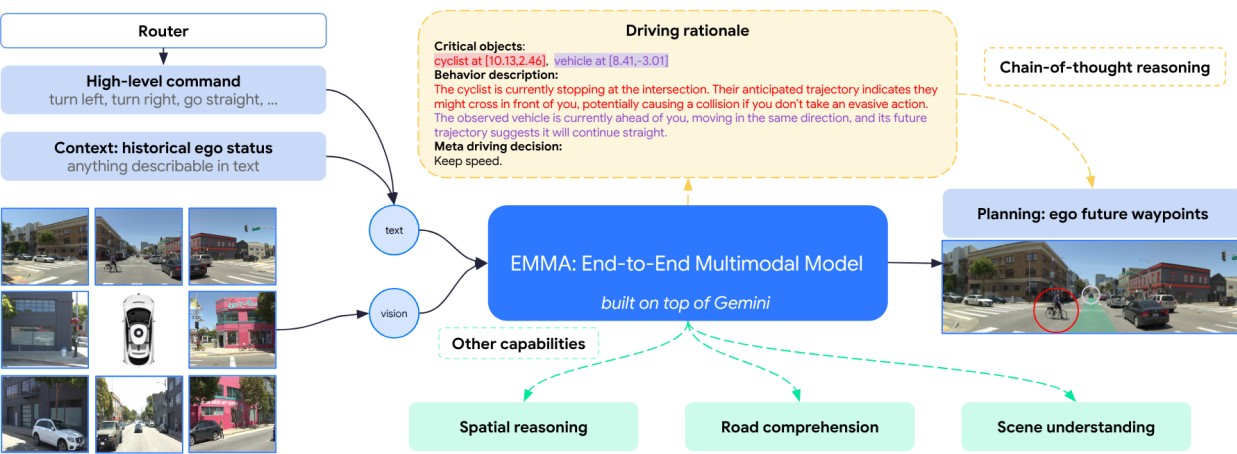

Figure 1: EMMA overview diagram. It takes 3 inputs (**left**): 1) a high-level command from the router, 2) historical status of the ego vehicle, and 3) surround-view camera videos. The model then predicts ego future trajectories (**right**) for motion planning that will be transformed into vehicle driving control signals. Further, we can ask the model to explain its rationale (**top**) before predicting trajectories, which enhances both the performance and explainability of the model through chain-of-thought reasoning. Notably, we incorporate visual grounding into the rationale so that the model also predicts the accurate 3D/BEV location for critical objects. In addition to end-to-end planning, we highlight three additional perception capabilities of our model (**bottom**).

## 2 Method

While we will show EMMA can be compatible with various MLLMs such as Gemini (Gemini Team Google, 2023) and PaLI (Chen et al., 2024d) in our experiments, this section will focus on our main EMMA based on Gemini. We leverage the auto-regressive Gemini models that are trained to process interleaved textual and visual inputs to produce text outputs:

$$\mathbf{O} = \mathcal{G}(\mathbf{T}, \mathbf{V}) \tag{1}$$

where $\mathcal{G}$ is the Gemini model, $\mathbf{O}$ represents natural language outputs, $\mathbf{T}$ represents natural language prompts, and $\mathbf{V}$ denotes images or videos. The language output $\mathbf{O} = (o_1, o_2, ..., o_n)$ is generated via next-token prediction, i.e., the output probability can be represented as $P(\mathbf{O}|\mathbf{T}, \mathbf{V}) = \prod_{i=1}^{n} P(o_i|o_{<i}, \mathbf{T}, \mathbf{V})$ for $n$ output tokens. Our goal is to adapt $\mathcal{G}$ for autonomous driving applications, thereby harnessing the world knowledge obtained during its pre-training phase.

As shown in Figure 1, we map autonomous driving tasks into our Gemini-based EMMA formulation. All sensor data are represented as stitched images or videos $\mathbf{V}$; all router commands, driving context, and task-specific prompts are represented in language prompts $\mathbf{T}$; and all output tasks are presented as language outputs $\mathbf{O}$. A challenge is that many of the inputs and outputs need to capture 3D world coordinates, such as waypoint BEV (Bird's Eye View) locations $(x, y)$ for motion planning and the location and size of 3D boxes. We consider two representations: The first is direct text conversion to floating-point numbers, expressed as $\mathbf{T}_{\text{coordinates}} = \{(x_i, y_i)\} \approx \text{text}(\{(x_i, y_i)\})$, where the specified decimal precision depends on the distance unit and decimal points. RT-2 (Brohan et al., 2023) exemplifies this approach in robotic control. The second approach uses special tokens to represent each location or action, formulated as $\mathbf{T}_{\text{coordinates}} = \{(x_i, y_i)\} \approx \text{tokenize}(\{(x_i, y_i)\})$, with resolution determined by the learned or manually defined discretization scheme. MotionLM (Seff et al., 2023) leverages this method for motion forecasting. We note that the two approaches have their respective strengths and weaknesses. We opt for the text representation such that all tasks can share the same unified language representation space and they can maximally reuse the knowledge from the pre-trained weights, even though the text representation may produce more tokens than specialized tokenization.

## 2.1 End-to-End Motion Planning

EMMA employs a unified, end-to-end trained model to generate future trajectories for autonomous vehicles directly from sensor data. These generated trajectories are then transformed into vehicle-specific control actions such as acceleration and turning for autonomous vehicles. EMMA's end-to-end approach aims to emulate human driving behavior, focusing on two critical aspects: (1) first, the use of navigation systems (e.g. Google Maps) for route planning and intent determination, and (2) second, the utilization of past actions to ensure smooth, consistent driving over time.

Our model incorporates three key inputs to align with these human driving behaviors:

1. **Surround-view camera videos** ($\mathbf{V}$): Provides comprehensive environment information.
2. **High-level intent command** ($\mathbf{T}_{\text{intent}}$): Derived from the router, includes directives such as *"go straight", "turn left", "turn right", etc.*
3. **Set of historical ego status** ($\mathbf{T}_{\text{ego}}$): Represented as a set of waypoint coordinates in Bird's Eye View (BEV) space, $\mathbf{T}_{\text{ego}} = \{(x_t, y_t)\}_{t=-1}^{-T_h}$ for $T_h$ timestamps. All waypoint coordinates are represented as plain text without specialized tokens. This can also be extended to include higher-order ego status such as velocity and acceleration.

The model generates future trajectories for motion planning, represented as a set of future trajectory waypoints for the ego vehicle in the same BEV space: $\mathbf{O}_{\text{trajectory}} = \{(x_t, y_t)\}_{t=1}^{T_f}$ for future $T_f$ timestamps, where all output waypoints are also represnted as plain text. Putting everything together, the complete formulation is expressed as:

$$\mathbf{O}_{\text{trajectory}} = \mathcal{G}(\mathbf{T}_{\text{intent}}, \mathbf{T}_{\text{ego}}, \mathbf{V}). \tag{2}$$

We then fine-tune Gemini with this formulation for end-to-end planner trajectory generation, as illustrated in Figure 1. We highlight 3 characteristics of this formulation:

1. **Self-supervised**: the only required supervision is the future locations of the ego vehicle. No dedicated human labels are needed.

2. **Camera-only**: the only sensor input required is surround-view cameras.

3. **HD map free**: no HD map is needed beyond the high-level routing information from a navigation system such as Google Maps.

While we are not the first to adopt this general formulation—(Li et al., 2024) conducted a thorough investigation, particularly examining the impact of including the historical ego status—our contribution lies in adapting this formulation specifically for MLLMs for autonomous driving. Our self-supervised approach exists alongside other notable methods that explore reconstruction via spatio-temporal scene decomposition (Yang et al., 2023), world modeling (Zhang et al., 2023a), or joint motion prediction (Wagner et al., 2024). Beyond this self-supervised foundation, the following sections explore enhancements for EMMA by incorporating reasoning and developing generalist setups with human or auto-labels.

## 2.2 Planning with Chain-of-Thought Reasoning

Chain-of-thought prompting (Wei et al., 2022) is a powerful tool in MLLMs that enhances reasoning capabilities and improves explainability. In EMMA, we incorporate chain-of-thought reasoning into end-to-end planner trajectory generation by asking the model to articulate its decision rationale $\mathbf{O}_{\text{rationale}}$ while predicting the final future trajectory waypoints $\mathbf{O}_{\text{trajectory}}$.

We structure the driving rationale hierarchically, progressing from 4 types of coarse-to-fine-grained information:

R1 - **Scene description** broadly describes the driving scenarios, including weather, day of time, traffic situations, and road conditions. We provide a concrete example for prompting Gemini. Example prompt: *Assume you are an autonomous vehicle, and the images come from your front cameras. Can you describe the current scenario in terms of weather, time of the day, road environment, lane options, and your ego lane position?*

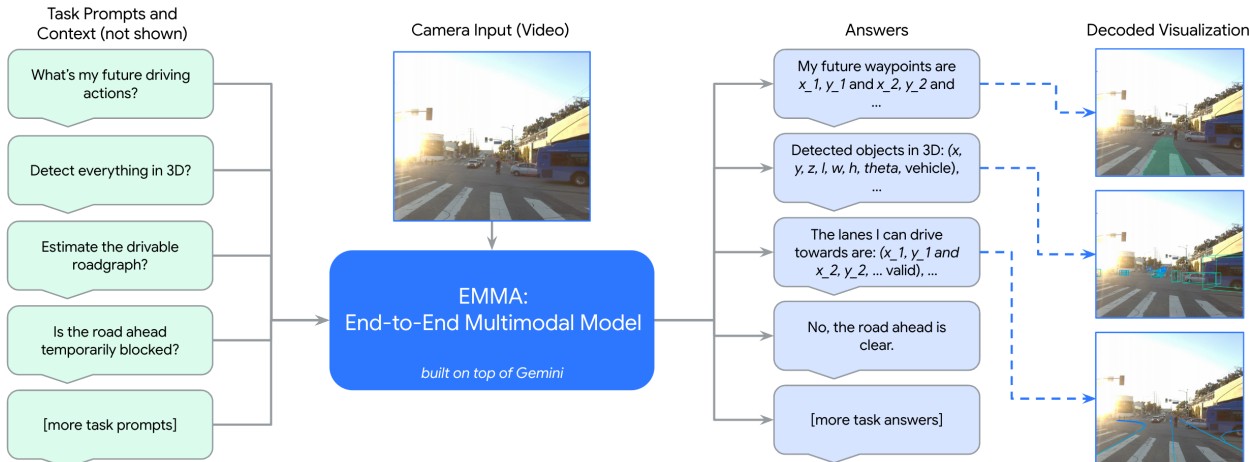

Figure 2: Illustration of EMMA Generalist. Starting with a task prompt (**left**), EMMA generates a corresponding textual prediction (**middle right**), which can then be decoded into a target output format, visualized and overlaid with the input image (**right**). EMMA Generalist is highly versatile, capable of performing a wide range of driving-related tasks, such as end-to-end motion planning, object detection, road graph estimation, and scene understanding Q&A. In the answer text, italicized words represent placeholders that will be dynamically substituted with actual values during task execution.

> Example answer: *The weather is clear and sunny, and it is daytime. The road is four-lane undivided street with a crosswalk in the middle. There are cars parked on both sides of the street.*

R2 - **Critical objects** are the on-road agents that can potentially influence the driving behavior of the ego vehicle, and we require the model to identify their precise 3D/BEV coordinates. For instance: *pedestrian at [9.01, 3.22], vehicle at [11.58, 0.35].*

R3 - **Behavior description of critical objects** describes the current status and intent for the identified critical objects. We provide a concrete example for prompting Gemini. Example prompt: *Assess the potential risks posed by the [focused_agent] with red bounding box. Summarize any immediate concerns that need addressing to maintain safety, paying close attention to how the objects may affect your route.* Example answer: *The pedestrian is currently standing on the sidewalk, looking toward the road, and maybe preparing to cross the street. The vehicle is currently ahead of me, moving in the same direction, and its future trajectory suggests it will continue straight.*

R4 - **Meta driving decision** includes 12 categories of high-level driving decisions, summarizing the driving plan given the previous observations. An example would be *I should keep my current low speed.*

We highlight that the driving rationale captions are generated using an automated tool without any additional human labels, ensuring scalability of the data generation pipeline. Specifically, we leverage off-the-shelf perception and prediction expert models to identify critical agents, and then use Gemini models with carefully designed visual and text prompts to generate scene and agent behavior descriptions. Meta driving decisions are computed using a heuristic algorithm that analyzes the ego vehicle's ground-truth trajectory.

During both training and inference, the model predicts all four components of the driving rationale before predicting the future waypoints, i.e.,

$$(\mathbf{O}_{\text{rationale}}, \mathbf{O}_{\text{trajectory}}) = \mathcal{G}(\mathbf{T}_{\text{intent}}, \mathbf{T}_{\text{ego}}, \mathbf{V}). \tag{3}$$

Where $\mathbf{O}_{\text{rationale}}$ denotes an ordered text output of (R1, R2, R3, R4) for driving rationale. Empirically, we observe that the prediction order of $\mathbf{O}_{\text{rationale}}$ and $\mathbf{O}_{\text{trajectory}}$ does not result in a significant difference in quality after model convergence. This suggests that we can predict $\mathbf{O}_{\text{trajectory}}$ first and apply early stopping during inference for time-critical applications.

### 2.3 EMMA Generalist

While end-to-end motion planning is the ultimate core task, a comprehensive autonomous driving system requires additional capabilities. Specifically, it must perceive the 3D world and recognize surrounding objects, the road graph and the traffic conditions. To achieve this goal, we formulate EMMA as a generalist model capable of handling multiple driving tasks through training mixtures.

Our vision-language framework represents all non-sensor inputs and outputs as plain text, providing the flexibility necessary to incorporate many other driving tasks. We employ instruction-tuning, a well-established approach in LLMs, to jointly train all tasks together with task-specific prompts included in the inputs $\mathbf{T}$ of Eq. 1. We organize these tasks into three primary categories: **spatial reasoning**, **road graph estimation**, and **scene understanding**.

**Spatial reasoning** is the ability to understand, reason, and draw conclusions about objects and their relationships in space. This enables an autonomous driving system to interpret and interact with its surrounding environment for safe navigation.

Our primary focus for spatial reasoning is **3D object detection**. We follow Pix2Seq (Chen et al., 2022a) and formulate the output 3D bounding boxes as $\mathbf{O}_{\text{boxes}} = \text{set}\{\text{text}(x, y, z, l, w, h, \theta, cls)\}$ where $(x, y, z)$ are the center location in the vehicle frame, $l, w, h$ are the length, width, and height of the box, $\theta$ is the heading angle, and $cls$ is the class label in text. We convert a 7D box to text by writing floating-point numbers with two decimal places, separated by spaces between each dimension.

We then represent the detection tasks using a fixed prompt $\mathbf{T}_{\text{detect\_3D}}$, such as "*detect every object in 3D*", as follows:

$$\mathbf{O}_{\text{boxes}} = \mathcal{G}(\mathbf{T}_{\text{detect\_3D}}, \mathbf{V}). \tag{4}$$

While $\mathbf{O}_{\text{boxes}}$ is an unordered set of boxes, the predictions from an auto-regressive language model are always ordered. We find that sorting the 3D bounding boxes by depth improves detection quality, unlike the findings in Pix2Seq (Chen et al., 2022a).

**Road graph estimation** focuses on identifying critical road elements for safe driving, including semantic elements (e.g., lane markings, signs) and physical properties (e.g., lane curvature). The collection of these road elements forms a road graph. For example, lane segments are represented by (a) nodes, where the lanes encounter an intersection, merge, or split and (b) edges between these nodes following the direction of traffic. The full road-graph is composed of many such polyline segments.

While edges within each polyline are directional, each polyline does not necessarily have a unique order relative to the other elements. This bears similarity to object detection (e.g., (Carion et al., 2020; Chen et al., 2022a)), where each box is defined by ordered attributes (top-left corner, bottom-right corner), but a relative ordering between boxes does not necessarily exist. There are several existing works that model polyline graphs with Transformers (Yuan et al., 2024; Liao et al., 2024a;b; 2023; Ding et al., 2023; Qiao et al., 2023; Liu et al., 2023; Li et al., 2022a), sharing similarities with language models.

Our general modeling formulation in EMMA is as follows:

$$\mathbf{O}_{\text{roadgraph}} = \mathcal{G}(\mathbf{T}_{\text{estimate\_roadgraph}}, \mathbf{V}). \tag{5}$$

where $\mathbf{O}_{\text{roadgraph}}$ is a text-encoded road graph represented as waypoints, $\mathbf{T}_{\text{estimate\_roadgraph}}$ is a prompt asking the model to predict the roadgrah, and $\mathbf{V}$ denotes the surrounding images.

We focus specifically on predicting drivable lanes, i.e., the lanes that the ego vehicle can drive towards in the scene. These are neighboring lanes in the same traffic direction and lanes branching out from the current ego lane. To construct $\mathbf{O}_{\text{roadgraph}}$, we **(a)** convert lanes into sets of ordered waypoints and **(b)** transform these sets of waypoints into text. It is beneficial to use sample-ordered waypoints to represent both traffic direction and curvature. Just like detection, we also find that ordering lanes by approximate distance improves the prediction quality. An example of our polyline text encoding is: `"(x1,y1 and...  and xn,yn);..."` where `"x,y"` are floating point waypoints with precision to 2 decimal places, `";"` separates polyline instances.

**Scene understanding** tasks test the model's understanding of the whole scene context, which can be relevant for driving. For example, roads can be temporarily obstructed due to construction, emergency situations, or other events. Detecting these blockages in a timely manner and safely navigating around them is essential for ensuring the smooth and safe operation of autonomous vehicles; however, multiple cues in the scene are required to determine if there is a blockage or not. We focus on how our model performs on this *temporary blockage detection* task, using the following formulation:

$$\mathbf{O}_{\text{temporary\_blockage}} = \mathcal{G}(\mathbf{T}_{\text{temporary\_blockage}}, \mathbf{T}_{\text{road\_user}}, \mathbf{V}), \tag{6}$$

where $\mathbf{O}_{\text{temporary\_blockage}}$ is the model output signaling potential obstruction, $\mathbf{V}$ denotes the surrounding images, $\mathbf{T}_{\text{road\_users}}$ denotes the all the objects on the road ahead, $\mathbf{T}_{\text{temporary\_blockage}}$ is the text prompt `"is the road ahead temporarily blocked?"`.

## 2.4 Generalist Training

Our unified vision-language formulation enables the simultaneous training of multiple tasks with a single model, allowing for task-specific predictions at inference time through simple variations of the task prompt $\mathbf{T}_{\text{task}}$. This training procedure is both straightforward and flexible.

For each task, we construct a dataset $\mathbf{D}_{\text{task}}$ containing $|\mathbf{D}_{\text{task}}|$ training examples. During each training iteration, we randomly sample a batch from the available datasets, with the probability of selecting an example from a specific dataset proportional to the dataset size: *i.e.*, $|\mathbf{D}_{\text{task}}|/\sum_t |\mathbf{D}_{\text{t}}|$. To train for $e$ epochs, we set the total number of training iterations to $e \times \sum_t |\mathbf{D}_{\text{t}}|$, ensuring that the training ratio among tasks is governed by the relative dataset sizes. The optimal training ratio is influenced by several factors, including task complexity, inter-task correlations, and the degree of transferability across tasks.

Our experimental results demonstrate that the generalist model, trained across multiple tasks, consistently outperforms each specialist model that is trained on a single task. This highlights the advantage of the generalist approach: enhanced knowledge transfer, improved generalization, and increased efficiency.

## 3 Experiments

Our experiments are primarily based on Gemini 1.0 Nano-1 (Gemini Team Google, 2023), with additional results provided for a variant of EMMA based on PaLI (Chen et al., 2024d). We first summarize the main datasets used for various experiments in Section 3.1. And then we present the results of end-to-end planner trajectory generation on two public datasets in Section 3.2. Next, we conduct experiments on our internal datasets, studying the impact of chain-of-thought and data scaling in Section 3.3. Section 3.4 focuses on 3D object detection experiments. Our co-training results for the generalist model are summarized in Section 3.5. Finally, we showcase visual results that highlight EMMA's capabilities in challenging, long-tail scenarios in Section 3.6.

### 3.1 Summary of Datasets

Before diving into the experimental details on how we validate EMMA, we summarize the main datasets in this section. Overall, we leverage three public datasets, nuScenes (Caesar et al., 2020), Waymo Open Motion Dataset (WOMD) (Chen et al., 2024a) and Waymo Open Dataset (WOD) (Sun et al., 2020). We also constructed three large-scale internal datasets for end-to-end motion planning, 3D detection and road graph estimation. We summarize the dataset sizes in Table 1.

The **nuScenes** dataset (Caesar et al., 2020) offers a comprehensive autonomous vehicle sensor suite for evaluation. It consists of 1,000 scenes, each spanning 20 seconds, and includes information from 6 cameras that collectively provide 360-degree coverage in the field of view.

The **WOMD** dataset comprises 103k real-world urban and suburban driving scenarios, each lasting 20 seconds. These scenarios are further segmented into 1.1M examples, each representing a 9-second window: 1

| Dataset Name | Total Hours of Driving | Number of Training Examples |
|---|---|---|
| nuScenes (Caesar et al., 2020) | 6 | 18,686 |
| WOMD (Chen et al., 2024a) | 572 | 487,061 |
| Internal Motion Planning Dataset | 203,117 **(355x)** | 24,374,046 **(50x)** |
| WOD (Sun et al., 2020) | 6 | 158,081 |
| Internal Detection Dataset | 6250 | 11,765,140 |
| Internal Roadgraph Dataset | 64,135 | 8,304,671 |

Table 1: Summary of main training dataset scales. This table details the scales of the three public datasets (nuScenes, WOMD, WOD) and three large-scale internal datasets leveraged for studying data scaling and generalist properties.

second is used as input context, and the remaining 8 seconds serve as the prediction target. The dataset includes detailed map features such as traffic signal states and lane characteristics, along with agent states such as position, velocity, acceleration, and bounding boxes.

We build a **large-scale internal motion planning dataset**, boasting over 24 million real-world driving scenarios, each 30 seconds long. This makes it roughly 355 times larger than WOMD or any other publicly available driving dataset. To efficiently leverage this massive scale, we sample just one frame per scenario, yielding 24 million diverse training examples. This approach maximizes dataset diversity while maintaining computational efficiency.

We also construct two separate, **large-scale internal datasets for detection and road graph tasks**, comprising 12 million and 8 million examples, respectively. For the detection dataset, we prioritized scenarios with diverse objects, sampling one example every 3 seconds from these scenarios. The road graph dataset, on the other hand, focuses on diverse scenarios and geo-locations, so we sample one example every 30 seconds, aligning with our motion-planning dataset.

Lastly, we also validate the camera-based 3D object detection task on the public **WOD** benchmark (Sun et al., 2020; Hung et al., 2024). This benchmark offers 1150 20-second scenes, each providing meticulously synchronized and calibrated high-quality LiDAR, camera, and 3D box data from a variety of urban and suburban environments.

### 3.2 End-to-End Motion Planning on Public Datasets

We conduct the end-to-end planner trajectory generation experiments on two public datasets, WOMD (Chen et al., 2024a) and the nuScenes dataset (Caesar et al., 2020). EMMA is trained with the simplest end-to-end planner trajectory generation formulation as in Equation 2, unless specified otherwise. That is, given camera images, ego vehicle history, and driving intent, the model is asked to predict the future ego waypoints for a certain time horizon.

#### 3.2.1 Driving on the Waymo Open Motion Dataset (WOMD)

For fair comparisons, we align our settings with WOMD. Additionally, we reproduce and adapt internally enhanced versions of the state-of-the-art motion prediction models, MotionLM (Seff et al., 2023) and Wayformer (Nayakanti et al., 2023), serving as our planner baselines. These baseline models are augmented with high-level command intent as an additional input.

As shown in Table 2, our model outperforms the MotionLM baseline when we train on the same dataset, with Gemini pre-trained weights. When pre-trained with our mega-scale internal dataset (denoted as EMMA+), our model outperforms both MotionLM and Wayformer. The full EFM+ (w/ CoT) surpasses the previous state-of-the-art models significantly by 13.5% at the 5s prediction horizon. We also apply the EMMA method to an open-sourced MLLM, PaLI-X (Chen et al., 2024d), denoted as EMMA[†] (PaLI). We show that EMMA

| Method | L2 (m) 1s | L2 (m) 3s | L2 (m) 5s |
|---|---|---|---|
| MotionLM* (Seff et al., 2023) | 0.045 | 0.266 | 0.696 |
| Wayformer* (Nayakanti et al., 2023) | 0.046 | 0.252 | 0.628 |
| EMMA$^{\dagger}$ (based on PaLI) | 0.034 | 0.274 | 0.797 |
| EMMA+$^{\dagger}$ (based on PaLI) | 0.031 | 0.239 | 0.680 |
| EMMA | 0.032 | 0.248 | 0.681 |
| EMMA (w/ CoT) | 0.030 | 0.241 | 0.664 |
| EMMA+ | 0.030 | 0.225 | 0.610 |
| EMMA+ (w/ CoT) | **0.027** | **0.203** | **0.543** |

Table 2: End-to-end motion planning experiments on an internal planning benchmark. CoT denotes equipping with chain-of-thought reasoning (Eq. 3). EMMA+ achieves the best quality across different prediction time horizons. EMMA$^{\dagger}$ and EMMA+$^{\dagger}$ denotes using PaLI-X (Chen et al., 2024d) as our base model, while the default EMMA and EMMA+ use Gemini as the base model. *Enhanced, reproduced baselines.

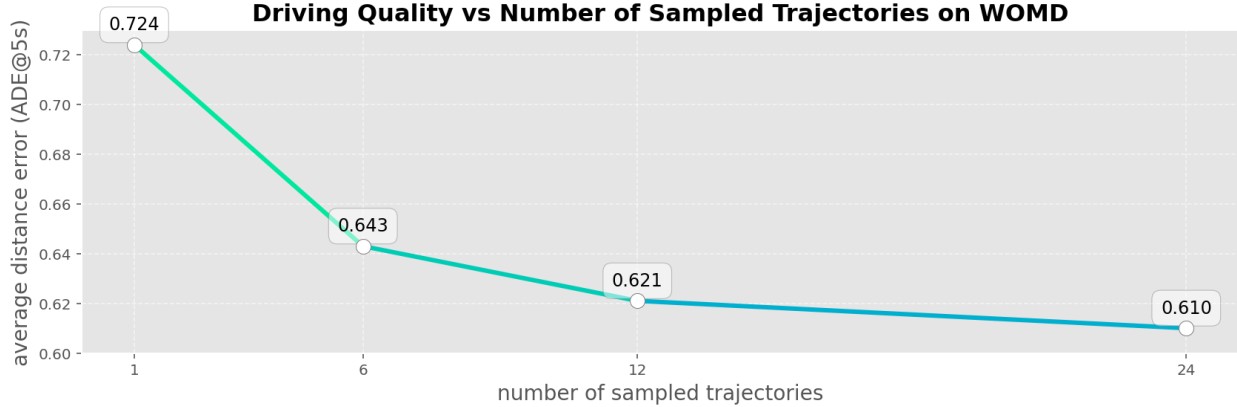

Figure 3: Ablation study on the number of sampled trajectories. As more trajectories are sampled, the quality measured by ADE@5s also improves, but the benefits diminish after 12+ samples.

can generalize well across different MLLMs with a large amount of training data, yielding better quality (at 1s and 3s) than previous state-of-the-art baselines.

We note the differences in inputs between MotionLM and EMMA: MotionLM takes inputs of agent location history, agent interactions, the road graph, and traffic light states. These agent boxes are produced by specialized off-board perception models that look at both past and future observations and are trained with a large amount of carefully curated human labels, the road graph is manually generated using full run segments, and all inputs heavily use LiDAR data with superior depth estimation. In stark contrast, EMMA only takes camera images and ego vehicle history as input, without the need of any labels or additional models (besides leveraging the Gemini pre-trained weights).

During inference, sampling a final trajectory from multiple candidates plays a critical role in the final performance. Both MotionLM and Wayformer generate 192 candidate trajectories, which are subsequently aggregated into 6 clusters using k-means clustering, resulting in 6 representative trajectories to be selected as the final output according to their probabilities. For fairness, we also sample multiple trajectories using a Top-$K$ decoding strategy, up to $K = 24$. We then compute the pairwise L2 distance between all trajectories and select the one with the lowest average L2 distance as the final predicted trajectory, which can be viewed as the "median" trajectory among all the predictions. We investigate the impact of the number of sampled trajectories on ADE, as illustrated in Figure 3. The results highlight that sampling from multiple trajectories leads to a notable improvement in ADE, however, with diminishing return.

| Method | self-supervised? | L2 (m) 1s | L2 (m) 2s | L2 (m) 3s | Avg L2 (m) |
|---|:---:|:---:|:---:|:---:|:---:|
| UniAD (Hu et al., 2023) | ✗ | 0.42 | 0.64 | 0.91 | 0.66 |
| DriveVLM (Tian et al., 2024) | ✗ | 0.18 | 0.34 | 0.68 | 0.40 |
| VAD (Jiang et al., 2023) | ✗ | 0.17 | 0.34 | 0.60 | 0.37 |
| OmniDrive (Wang et al., 2024a) | ✗ | 0.14 | 0.29 | 0.55 | 0.33 |
| Ego-MLP* (Zhai et al., 2023) | ✓ | 0.15 | 0.32 | 0.59 | 0.35 |
| BEV-Planner (Li et al., 2024) | ✓ | 0.16 | 0.32 | 0.57 | 0.35 |
| EMMA (random init) | ✓ | 0.15 | 0.33 | 0.63 | 0.37 |
| EMMA | ✓ | 0.14 | 0.29 | 0.54 | 0.32 |
| EMMA+ | ✓ | **0.13** | **0.27** | **0.48** | **0.29** |

Table 3: End-to-end motion planning experiments on nuScenes (Caesar et al., 2020). EMMA (random init) denotes models are randomly initialized; EMMA denotes models are initialized from Gemini; EMMA+ denotes models that are pre-trained on our mega-scale internal data. EMMA achieves state-of-the-art performance on the nuScenes planning benchmark, outperforming the supervised (with perception and/or human labels) prior art by 6.4% and self-supervised (no extra labels) prior art by 17.1%. *Ego-MLP results are taken from a reproduced version in BEV-Planner.

### 3.2.2 Driving on the nuScenes Dataset

In our experiments, we follow the standard protocol of nuScenes for planning evaluation: predict the next 3 seconds of future driving actions based on 2 seconds of historical data. We measure the planning quality with L2 errors at 1-, 2- and 3-second time horizons, aligning with established baseline methods, in particular BEV-Planner (Li et al., 2024).

We train and evaluate EMMA with the simplest end-to-end planner trajectory generation formulation as in Equation 2 (self-supervised, without chain-of-thought reasoning nor generalist training). As shown in Table 3, our self-supervised EMMA achieves state-of-the-art results in planning on nuScenes, outperforming all previous supervised (with intermediate perception labels and/or human labels) and self-supervised (no extra labels) methods. Under the same self-supervised setup, EMMA outperforms BEV-Planner (Li et al., 2024) by 17.1% in average L2 metric; even compared to OmniDrive (Wang et al., 2024a) that heavily uses intermediate perception human labels, our self-supervised EMMA improves the average L2 metric by 12.1%.

Unlike in WOMD, we note that sampling multiple trajectories did not yield significant improvements. We hypothesize that this is due to nuScenes' shorter prediction time horizon (3s) in simpler driving scenarios. Thus, we report only top-1 predictions for our results.

### 3.3 End-to-End Motion Planning with Chain-of-Thought Reasoning on Internal Dataset

In this section, we present our studies of end-to-end planning with chain-of-thought on our internal dataset. This dataset contains 24 millions of scenarios, orders of magnitude larger than any publicly available autonomous driving dataset. The model takes in 2 seconds of history to predict the driving actions for 5 seconds into the future.

Table 4 presents the results of our experiments on chain-of-thought reasoning applied to end-to-end planning. By adopting the chain-of-thought formulation (Equation 3), we achieve a notable 6.7% improvement over the standard end-to-end planning approach detailed in Equation 2. We also conduct an ablation study to analyze the contributions of different rationale components. Our findings reveal that both driving meta-decision and critical object identification significantly enhance performance, contributing improvements of 3.0% and 1.5%, respectively. When these components are combined, the gains are even more substantial. Conversely, while scene description has a neutral impact on driving performance, it enhances the model's explainability. These results demonstrate that chain-of-thought reasoning can meaningfully improve driving performance, particularly when its components are carefully selected and integrated.

| Scene description | Critical object | Meta decision | Behavior description | Relative improvements over baseline e2e planning |
|:---:|:---:|:---:|:---:|:---:|
| ✓ | ✗ | ✗ | ✗ | + 0.0% |
| ✗ | ✓ | ✗ | ✗ | + 1.5% |
| ✗ | ✗ | ✓ | ✗ | + 3.0% |
| ✗ | ✓ | ✓ | ✗ | + 5.7% |
| ✗ | ✓ | ✓ | ✓ | + 6.7% |

Table 4: Ablation study on chain-of-thought reasoning components. It improves end-to-end planning quality by up to 6.7% by combining all elements. In particular, driving meta-decision and critical objects contribute the improvements of 3.0% and 1.5%, respectively. The details of each component is described in Section 2.2.

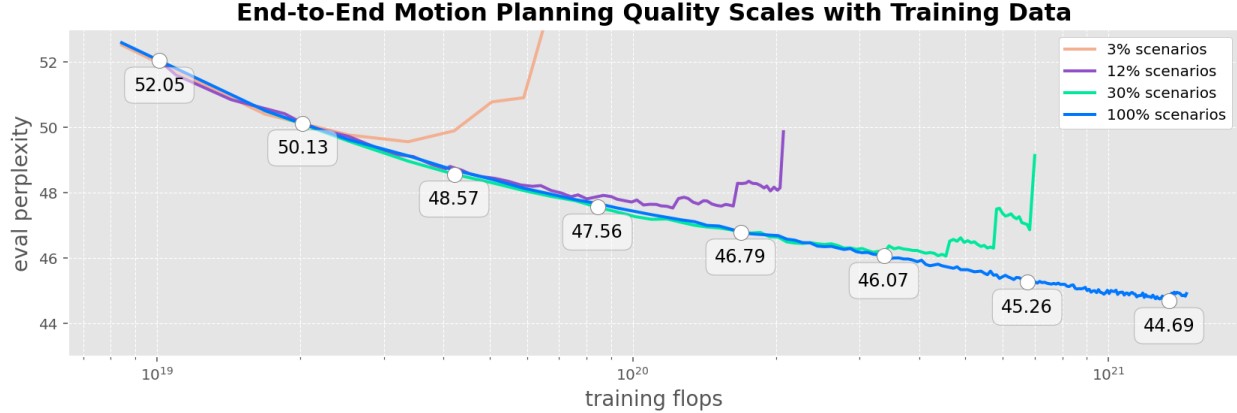

Figure 4: EMMA data scaling experiments on our mega-scale internal dataset. Each curve represents the eval perplexity for end-to-end motion planning as training proceeds with more steps. The x-axis is training compute, measured by floating-point operations (FLOPs) in log scale. The same EMMA model is trained on four sizes of datasets that are sampled with different percentages from 3% to 100% (denoted by different colors). In general, EMMA tends to achieve better quality until overfitting when given more training compute, but it overfits quickly on smaller datasets. We observe the driving quality has not saturated when using the full large-scale dataset.

We also perform a series of **data scaling** experiments for end-to-end planning, the results of which are illustrated in Figure 4. As we train the model on a larger training set, we observe lower eval perplexities before overfitting. Our findings indicate that the driving quality of EMMA has not yet plateaued, even with the current mega-scale dataset.

### 3.4   3D Object Detection

We validate our 3D object detection performance on the 3D camera-primary detection benchmark from the Waymo Open Dataset (Sun et al., 2020) using the Longitudinal Error Tolerant (LET) matching (Hung et al., 2024). We evaluate two versions: EMMA and EMMA+, similar to earlier sections, where EMMA+ is pre-trained on the 3D detection task using our internal dataset. The quantitative results are reported on the official test set and summarized in Figure 5.

Our findings show that after pre-training, EMMA+ achieves competitive performance on the benchmark. Since our model produces a set of detected boxes without individual confidence scores, we compare the precision/recall instead of LET-3D-AP, which is calculated based on the precision/recall curve. We also compare the commonly used F1-score, where EMMA's F1-score is computed using the single precision/recall and other models' F1-scores are calculated by picking the maximal F1-score on the curve (often called F1-max).

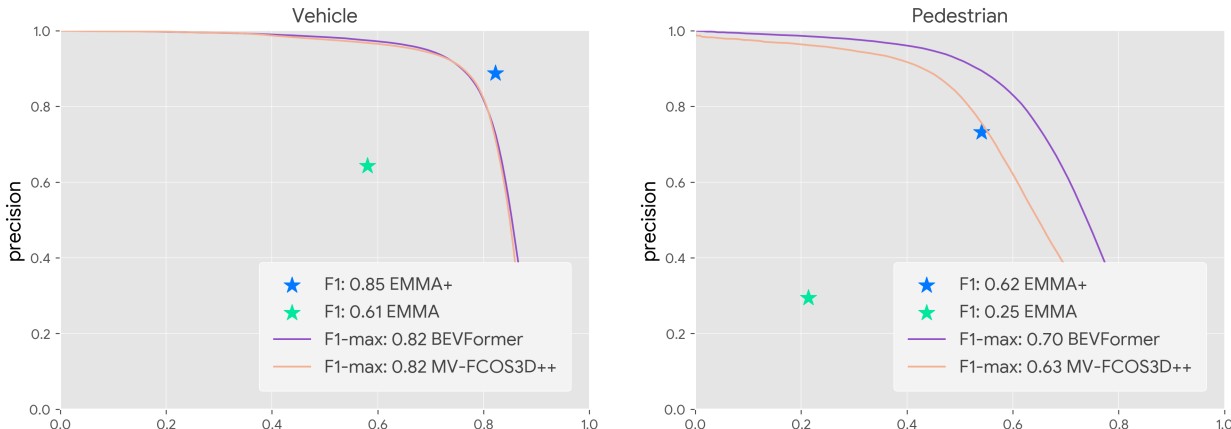

Figure 5: Camera-primary 3D object detection experiments on WOD (Sun et al., 2020) using the standard LET matching (Hung et al., 2024). EMMA+ achieves competitive performance on the detection benchmark in both precision/recall and F1-score metrics. Compared to state-of-the-art methods, it achieves 16.3% relative improvements in vehicle precision at the same recall or 5.5% recall improvement at the same precision.

Figure 5 shows the performance comparison. In generally, EMMA+ demonstrates substantial improvements over state-of-the-art methods such as BEVFormer (Li et al., 2022b), achieving a 16.3% relative increase in vehicle precision at the same recall, and a 5.5% recall improvement at the same precision. EMMA+ also achieve better F1-score than prior arts. Performance on the pedestrian class is also comparable to that of MV-FCOS3D++ (Wang et al., 2021). Additionally, we provide a performance breakdown across different ranges, highlighting that our model performs especially well in the near range. Our results underscore that with sufficient data and a large enough model, a multimodal approach can surpass specialized expert models in 3D detection quality.

### 3.4.1 Road Graph Estimation

Road graph estimation is a complex task that predicts a group of unordered polylines, each of which is represented as a sequence of waypoints. We measure the quality of road graph prediction with two metrics: (1) **lane-level** precision and recall, where we define a true positive match between a predicted lane polyline and a groundtruth lane polyline if and only if their Chamfer distance is within 1 meter; and (2) **pixel-level** precision and recall, where polylines are rasterized into a BEV grid with 1 meter resolution – we then treat the BEV grid as a image and compute precision and recall based on per-pixel matching.

As discussed in Section 2.3, this task involves several design choices. One is about the representation of road graph polylines, where our choice is to define the start and end points of each lane, with intermediate points added as needed to accurately capture the road's curvature. Another critical design choice is the construction of target label sequences used for model training. Drawing inspiration from Pix2Seq (Chen et al., 2022a) in the context of object detection, one effective design choice is to pad the targets and apply random shuffling. This technique helps the model handle unordered outputs and prevents premature termination during training.

Figure 6 presents our ablation studies on various design choices. Starting from our best designs, we systematically ablate each of the following configurations and assess the resulting quality degradation. We then summarize the key insights from our analysis.

**Polyline representation: dynamic sampling is better than fixed sampling**. A simple polyline representation is to sample a fixed number of sparse control points per lane, e.g. two end points plus a fixed number of intermediate points to capture curvature. However, we find a better approach is to dynamically adjust the number of points per polyline according to the curvature and length of the lane. By keeping a consistent waypoint density rather than a consistent number of waypoints, we achieve a representation

that more accurately captures the lane structure intricacies, yielding around a 40% to 90% difference in the metrics as shown in Figure 6. By adapting the waypoint density to the road geometry, particularly in areas with sharper curves or varying lane lengths, we achieve a flexible representation that more accurately captures the lane structure intricacies

**Polyline representation: ego-origin aligned sample intervals are better than naively aligned sample intervals**. The road graph is typically stored and accessed in global coordinate frame, meaning lane origins and extensions are independent of the ego vehicle position. To improve accuracy, it is essential to adjust lane point samples to start from the ego vehicle coordinate frame origin. Specifically, sampling polyline points relative to the AV position (ego-origin) avoids arbitrary offsets that can arise from directly transforming points sampled in the global coordinate frame into the ego coordinate frame. This prevents a 25% to 60% drop in prediction quality.

**Target sequence construction: shuffled ordering is better than arbitrary ordering**. We organize polyline targets into bins based on their endpoint distance from the ego vehicle, providing a rough global ordering. For instance, we categorize lanes into nearby lanes and those further away that serve as connecting lanes. During training, we dynamically shuffle the polylines within each distance bin to enhance the model robustness and coverage. This dynamic shuffling within each bin improves the model's ability to generalize across different lane configurations, leading to more accurate predictions.

**Target sequence construction: padding is better than non-padding**. Similar to Pix2Seq (Chen et al., 2022a), we find that padding targets to prevent early termination is highly effective. In addition to padding the total number of polyline targets, we also pad the number of points within each polyline. We use *"invalid"* tokens to represent padded points within polylines. Each polyline is also explicitly tagged with a final *"valid"* or *"invalid"* token to denote whether it contains any nonpadded points. This approach ensures consistent input sizes, which helps maintain the integrity of the model during training and reduces the risk of premature truncation, leading to more reliable and accurate predictions.

**Target sequence construction: adding punctuation and other semantically redundant token improves quality**. In the target sequence construction, we notice that it is beneficial to use language-like structures and punctuation to group targets (e.g., `"(x,y and x,y);..."` instead of `"xy xy;..."`). Additionally, explicitly including semantically redundant tokens – such as marking padded targets as *"invalid"* instead of relying on implicit omissions of *"valid"* markers – improves performance. This approach, incorporating punctuation and redundancy, results in a boost of up to 10% in lane-level metrics. We attribute this improvement to the language-related pre-training of Gemini. By leveraging similar structured expressions, Gemini can be more easily adapted to other tasks.

### 3.4.2 Scene Understanding

Figure 7 summarizes our studies on the scene understanding task for *temporary blockage detection*. Our study is based on our internal datasets specifically curated for these scenarios. For this study, we establish our baselines by showing a picture to human and asking them to judge whether a lane is temporarily blocked. They can answer 'yes', 'no', or 'unsure'. Our `baseline` will treat all 'unsure' answers as incorrect, `baseline+filtering` will filter out all examples with 'unsure' answers. In contrast, our model is fine-tuned to predict 'yes' or 'no' for all examples. As shown in the figure, our naive model that is directly fine-tuned for only this task achieves better performance than the baseline comparison, but underperforms on the baseline+filtering comparison. To boost the model performance, our first attempt is to co-train this task with road graph estimation, but the naive mixture doesn't improve performance. Our second attempt is to first pre-train the model on road graph estimation, and then fine-tune on these two tasks. Results show when the pre-training is long enough, the quality is boosted, showcasing the model's ability to integrate multiple tasks for enhanced performance.

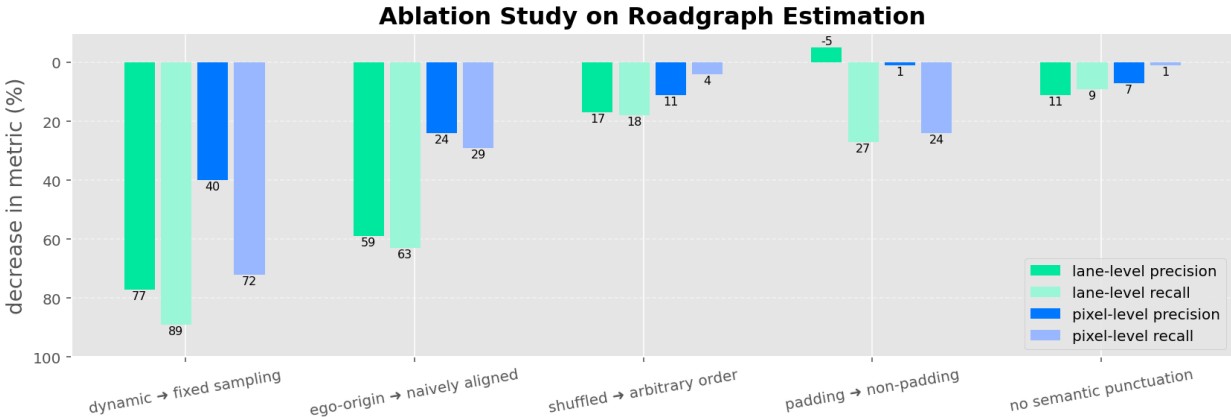

Figure 6: Ablation study on road graph estimation. To evaluate the influence of different components in our road graph estimation model, we ablate each configuration and measure the corresponding impact on quality. Dynamic sampling (**leftmost**) of road graph polylines based on lane curvature and length proves to be the most significant factor, leading to a substantial 70% to 90% change in lane-level precision and recall. In contrast, aligning the model with a language-like representation, *i.e.*, semantic punctuation (**rightmost**), has a more modest effect, contributing to only <10% change in precision and recall of any metric.

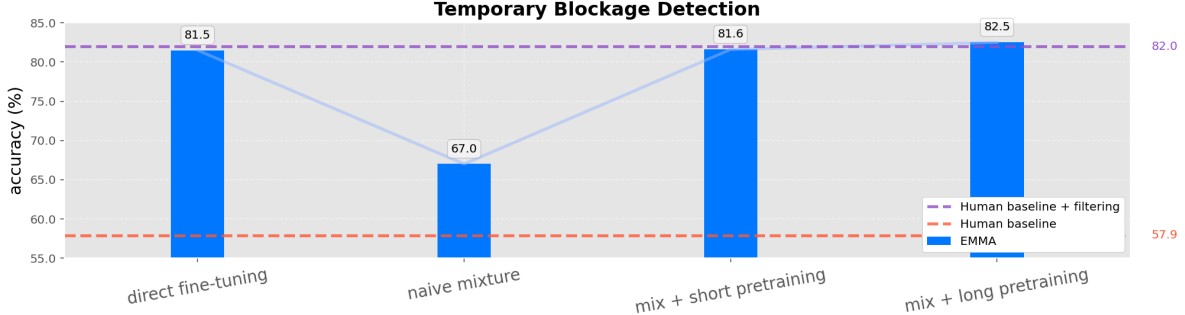

Figure 7: Scene understanding experiments. `direct fine-tuning` denotes solely using the temporal blockage data during fine-tuning; `naive mixture` denotes co-training this scene task with road graph estimation; `mix + short pretraining` denotes pre-training on road graph esitmation first, and then fine-tune on the mixture of both tasks; `mix + long pretraining` denotes a longer pre-training before fine-tuning. The naive fine-tuning is already close to strong human baseline, but long-pretraining with training mixture can further boost the quality.

## 3.5 Generalist

We explore the development of the EMMA Generalist by co-training on multiple tasks and analyzing their synergies, as summarized in Table 5. For this study, we focus on three core tasks: end-to-end planning, 3D object detection, and road graph estimation.

Co-training on all three tasks yields significant improvements, with the generalist model outperforming the single-task models by up to 5.5%. We attribute these results to the complementary nature of the tasks. For example, road graph estimation becomes easier when the model can accurately identify the locations of vehicles. Similarly, driving quality is closely tied to understanding agent interactions, a skill enhanced by 3D object detection. Paring the mixture down to only two tasks still yields improvements, with certain combinations leading to greater gains than others. For instance, detection performance improves most when co-trained with driving, and road graph estimation similarly benefits most when paired with driving. This

| e2e planning | 3D detection | road graph | Relative improvement over single task | | |
|---|---|---|---|---|---|
| | | | e2e planning | detection | road graph |
| | ✓ | ✓ | - | +1.6% (±1.0%) | +2.4% ( ±0.8%) |
| ✓ | ✓ | | +1.4% (±2.8%) | +5.6% (±1.1%) | - |
| ✓ | | ✓ | −1.4% (±2.9%) | - | +3.5% (±0.9%) |
| ✓ | ✓ | ✓ | +1.4% (±2.8%) | +5.5% (±1.1%) | +2.4% (±0.8%) |

Table 5: Generalist co-training experiments. (±∗) indicates standard deviation. By co-training on multiple tasks, EMMA gains a broader understanding of driving scenes, enabling it to handle various tasks at inference time, while enhancing individual task performance. Notably, certain task pairings yield greater benefits than others, suggesting these tasks are complementary. Co-training all three tasks together yields the best quality.

suggests the driving task plays a prominent and influential role, serving as a key contributor to overall performance improvements.

These findings suggest that pursuing a generalist model is a promising direction for future research, with the potential for deeper insights into task synergies and performance optimization.

### 3.6 Visualizations

We group visual examples by scenario type: Examples (a)-(d) showcase how EMMA safely interacts with rare, unseen objects or animals on the road. Examples (e)-(f) feature EMMA navigating through construction areas. Examples (g)-(j) showcase EMMA following traffic rules at intersections with traffic lights or traffic controllers. Examples (k)-(l) highlight EMMA respecting vulnerable road users like motorcyclists.

Given these examples, we demonstrate the following capabilities of EMMA:

- **Generalizability**: Adapts well to diverse real-world driving scenarios across different environments and attends to objects beyond its fine-tuning categories, such as squirrels.

- **Predictive driving**: Proactively adjusts to the behavior of other road users for safe and smooth driving.

- **Obstacle avoidance**: Consistently adjusts trajectories to avoid obstacles, debris and blocked lanes.

- **Adaptive behavior**: Safely handles complex situations like yielding, construction zones, and following traffic control signals.

- **Accurate 3D detection**: Effectively identifies and tracks road agents, including vehicles, cyclists, motorcyclists, and pedestrians.

- **Reliable road graph estimation**: Accurately captures road layouts and integrates them into safe trajectory planning.

To conclude, these scenarios highlight EMMA's capability to operate safely and efficiently in a variety of challenging and diverse driving scenarios and environments.

## 4 Related Works

**End-to-end autonomous driving research** enjoys a rich history and has evolved significantly since ALVINN (Pomerleau, 1988) employed shallow neural networks to predict control signals. The field benefited from further deep learning advancements: e.g. DAVE-2 (Bojarski et al., 2016) and ChauffeurNet (Bansal et al., 2019) leveraged deeper neural architectures and incorporated sophisticated perception and motion planning modules respectively. Recent research has expanded to include multimodal inputs (Codevilla et al., 2018; Prakash et al., 2021), multi-task learning (Chitta et al., 2022; Wu et al., 2022), reinforcement learning (Chekroun et al., 2023; Chen et al., 2021; Kendall et al., 2019; Liang et al., 2018; Toromanoff et al., 2020), and distillation (Chen et al., 2020; Zhang & Ohn-Bar, 2021; Zhang et al., 2021). Unified planning

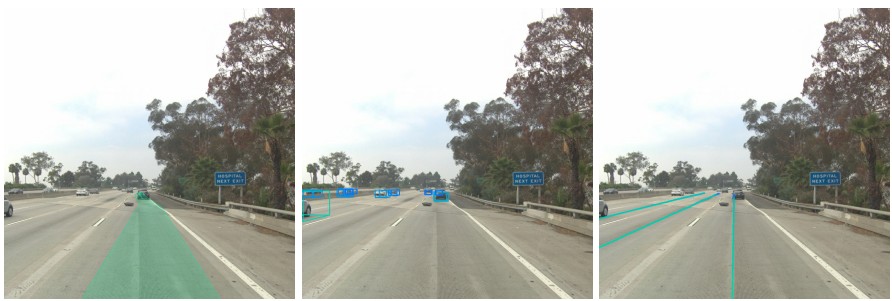

(a) A garbage bag appears on the freeway, so our predicted trajectory suggests to nudge slightly to the right to avoid it.

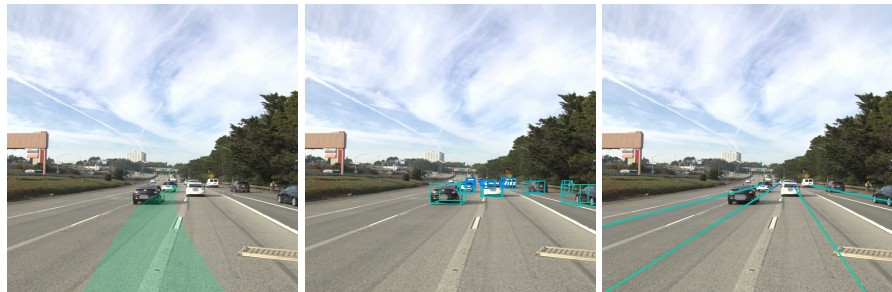

(b) A ladder appears on the freeway, and our predicted trajectory suggests to switch to the left lane to bypass it appropriately.

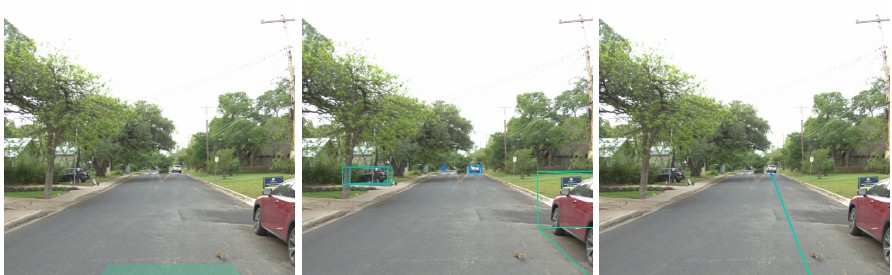

(c) we encounter a small squirrel on the road and our predicted trajectory instinctively slows down to avoid the animal. Note EMMA wasn't explicitly trained to detect squirrels.

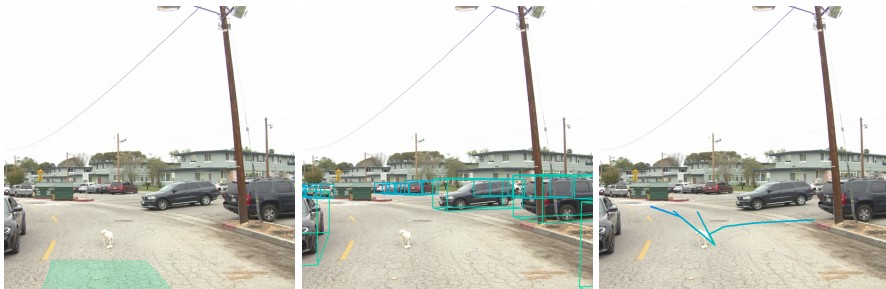

(d) A white dog appears in our lane, and our model predicts to slow down and yield. Our model also accurately detects surrounding vehicles, including those in adjacent lanes and the parking lot.

Figure 8: EMMA prediction visualization. Each row contains a scenario with our model's predictions: end-to-end planning (left), 3D object detection (middle), and road graph estimation (right).

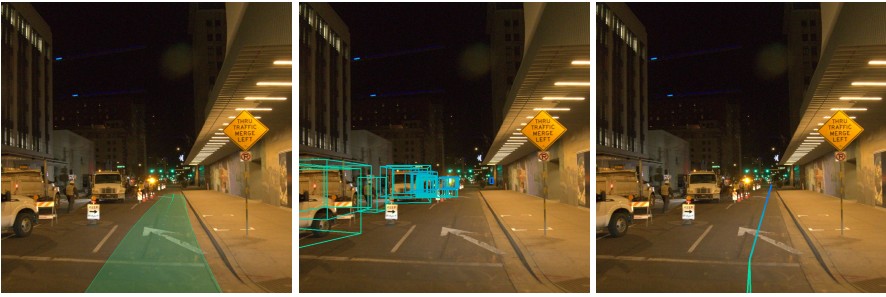

(e) As a construction zone blocks the left lanes, our predicted trajectory suggests passing through on the right, while the road graph estimation correctly identifies the blocked area.

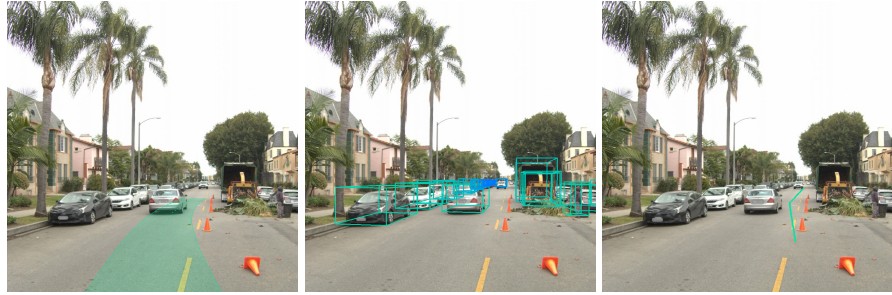

(f) Our lane is blocked by construction cones, so our predicted trajectory suggests to move into the left lane, even though it's in the opposite direction. EMMA captured the blockage and performed a detour.

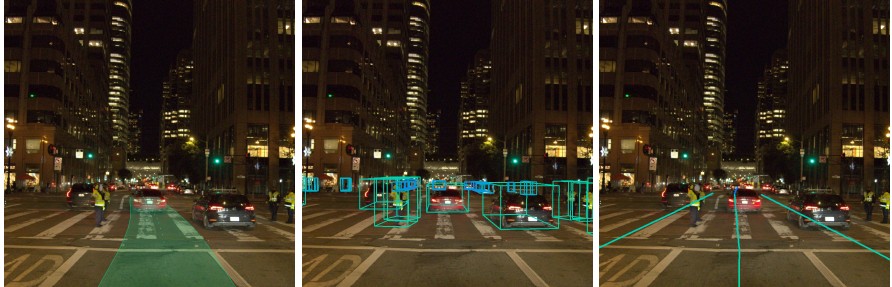

(g) A traffic controller signals to proceed through the intersection, and our predicted trajectory aligns with the instruction.

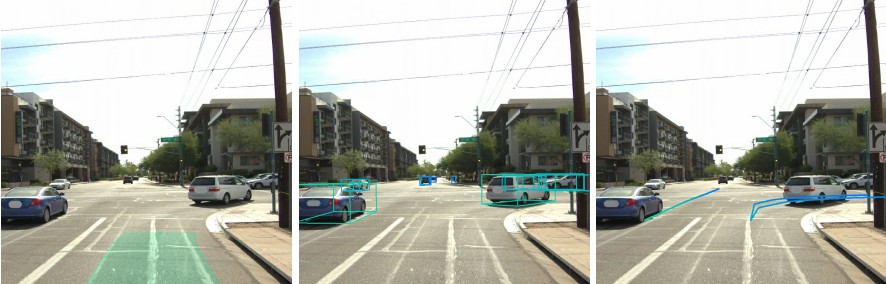

(h) Our predicted trajectory suggests to stop as we approach an intersection with a yellow light, demonstrating cautious and safe behavior.

Figure 9: EMMA prediction visualization. Each row contains a scenario with our model's predictions: end-to-end planning trajectory (left), 3D object detection (middle), and road graph estimation (right).

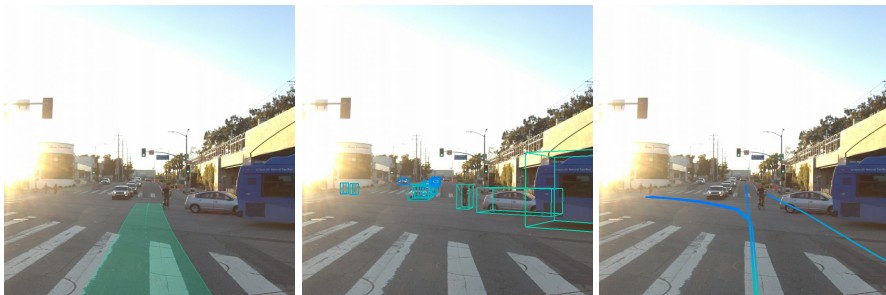

(i) While crossing an intersection, our predicted trajectory nudges slightly to the left due to nearby cars and a bicyclist partially occupying our lane.

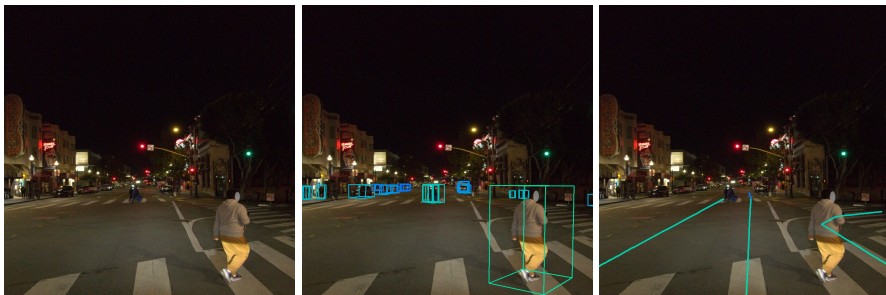

(j) Our model predicts a driving trajectory to patiently wait at a red light (left). The model also accurately predicts surrounding 3D objects (middle) and road graph with lane centers (right).

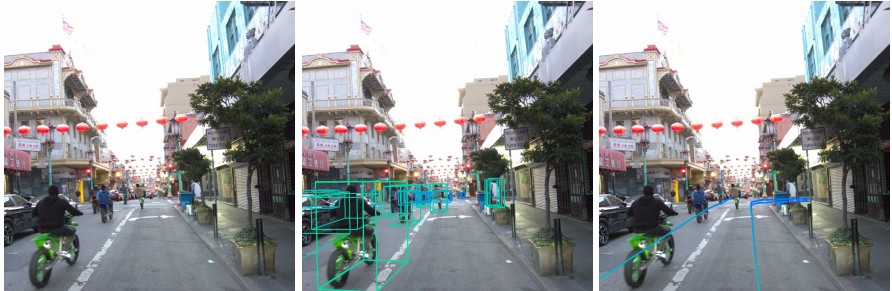

(k) A fleet of fast-moving motorcyclists pass by. The predicted trajectory suggests pausing to allow them to pass safely. Notably, motorcyclists are accurately identified by our model (middle).

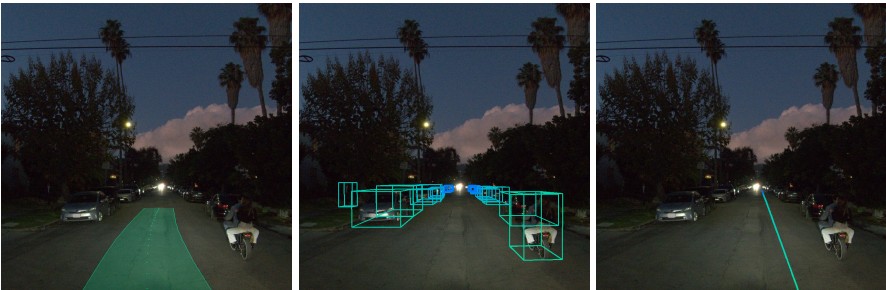

(l) A motorbike is moving on a narrow lane at night, and yields to the right. Our predicted trajectory adjusts, guiding us to pass safely by nudging slightly to the left.

Figure 10: EMMA prediction visualization. Each row contains a scenario with our model's predictions: end-to-end planning trajectory (left), 3D object detection (middle), and road graph estimation (right).

frameworks such as VAD (Jiang et al., 2023; Chen et al., 2024c), UniAD (Hu et al., 2023), PARA-Drive (Weng et al., 2024), and GenAD (Zheng et al., 2024) integrated planning with conventional modules in open-loop environments. More studies have been proposed to examine the robustness, safety, and transferability from synthetic environments to real-world domains. However, recent findings from AD-MLP (Zhai et al., 2023) and BEV-Planner (Li et al., 2024) revealed that these methods could potentially overfit to ego status despite their good performance on benchmarks. Our work revisits the simplicity of earlier end-to-end models such as ALVINN and DAVE-2, enhancing them with powerful MLLMs.

**Vision language models for autonomous driving** have gained increasing interest, focusing on achieving explainable driving behavior and generalizability through end-to-end learning frameworks. DriveGPT4 (Xu et al., 2024) and LMDrive Shao et al. (2024) utilize LLMs to explain vehicle actions and predict control signals in an iterative Q&A format. Drive Anywhere (Wang et al., 2024c) introduces patch-aligned feature extraction from MLLMs for text-based driving decision queries, while OmniDrive (Wang et al., 2024a) features a 3D vision-language model design for reasoning and planning. Other approaches use MLLMs in graph-based VQA contexts ((Sima et al., 2024)), integrate LLMs in a BEV-based planner ((Pan et al., 2024), or apply chain-of-thought reasoning ((Tian et al., 2024; Wang et al., 2024b; Bhattacharyya et al., 2023)) to tackle multiple driving-related tasks. Modular architectures such as LLM-Drive (Chen et al., 2024b) leverage LLMs with object-level vector inputs for planning. In contrast, our work studies end-to-end fine-tuning of a state-of-the art MLLM for driving tasks, employing a generalist approach that emphasizes open-world driving capabilities.

**Multimodal large language models** (MLLM) extend LLMs (Vaswani et al., 2017; Devlin, 2019; Raffel et al., 2020; Gemini Team Google, 2023; Reid et al., 2024; Chowdhery et al., 2023; Anil et al., 2023; Radford et al., 2018; 2019; Brown et al., 2020; Achiam et al., 2023; Touvron et al., 2023a;b; Dubey et al., 2024) to multiple modalities, leveraging their generalizability, reasoning capabilities, and contextual understanding. Early explorations (Donahue et al., 2015; Vinyals et al., 2015; Chen et al., 2022a) focused on specific vision-language problems or open-set object detection Liu et al. (2024b); Zareian et al. (2021); Gu et al. (2022), while recent research has scaled up both trask diversity and model sizes for improved generalizability and few-shot capabilities (Cho et al., 2021; Chen et al., 2022b; Wang et al., 2022; Lu et al., 2022; Alayrac et al., 2022; Yu et al., 2022; Chen et al., 2023; 2024d; Wang et al., 2024d; Peng et al., 2024; Huang et al., 2023; Lu et al., 2024). Notable examples include Flamingo (Alayrac et al., 2022), a 70B model (Hoffmann et al., 2022) which achieved state-of-the-art quality for multiple few-shot vision benchmarks, and CoCa (Yu et al., 2022) a 2.1B parameter model which demonstrated state-of-the-art performance on zero-shot transfer and various downstream tasks including ImageNet classification. PaLI (Chen et al., 2023; 2024d), at 55B parameters, achieves better performance across multiple vision and language tasks by scaling both the vision and language model components jointly. These early works demonstrate the strong performance and generalizability of MLLMs. Recent trends have seen the integration of native multi-modal inputs in LLMs, such as Gemini (Gemini Team Google, 2023; Reid et al., 2024), GPT-4o, and Llama3-v (Dubey et al., 2024; Liu et al., 2024a). Notably, researchers also apply MLLMs to robotic navigation (Zhang et al., 2024; Sun et al., 2024) and manipulation (Brohan et al., 2023; Alakuijala et al., 2024; Wang et al., 2023). Our work explores the application of these promising new models for generalist end-to-end autonomous driving.

## 5  Conclusion

In this paper, we present EMMA, a Gemini-powered end-to-end multimodal model for autonomous driving. It treats Gemini as a first class citizen and recasts autonomous driving tasks as vision question answering problems to fit the paradigm of MLLMs, aiming at maximizing the utility of Gemini's world knowledge and its reasoning capability equipped with chain-of-thought tools. Unlike historical cascaded systems with specialized components, EMMA directly maps raw camera sensor data into various driving-specific outputs, including planning trajectories, perception objects, and road graph elements. All task outputs are represented as plain text and thus can be jointly processed in a unified language space through task-specific prompts. Empirical results show that EMMA achieves state-of-the-art or competitive results on multiple public and internal benchmarks and tasks, including end-to-end planning, camera-primary 3D object detection, road graph estimation, and scene understanding. We also demonstrate that a single co-trained EMMA can

predict multiple tasks, while matching or even super-passing the performance of individually trained models, highlighting its potential as a generalist model for autonomous driving.

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

# A Appendix

We organize this appendix section as follows:

1. We summarize all 12 categories in the meta decision in the chain-of-thought reasoning formulation in Table 6.

2. We supply details on the 3D object detection metrics.

3. To provide insights for future development, we present three distinct failure examples in Figure 12.

4. To facilitate reproduction, we provide an example of the concrete prompts used for EMMA Generalist and their corresponding model-predicted answers in Table 7.

5. Despite the promising results, we acknowledge the limitations of our work and propose directions for future research in Section A.5.

## A.1 Meta Decision in Chain-of-Thought Reasoning

We describe the chain-of-thought reasoning in Section 2.2 in the main paper. One important component is the meta decision, where we first partition the decision into 12 categories with heuristics and transform them into natural languages.

We summarize the 12 categories in Table 6. We use speed at 3 different future timestamps, *i.e.*, at future 0, 1, and 3 seconds, as the decision points for different categories. If we are able to identify the cause of speed changes, *e.g.*, due to traffic signs or critical objects, then we also append it to the description. We plan to explore more fine-grained meta decisions and reasoning in a future work.

## A.2 Distance Breakdowns of 3D Detection Metrics

We plot the distance breakdowns of the camera-primary 3D object detection experiments in Figure 11. We observe that the performance gap between EMMA+ and the baseline models diminishes as the distance to objects increases. We attribute this phenomenon to the potentially lower resolution of the camera input used by EMMA compared to that of the baseline models.

| Speed at 0s | Speed at 1s | Speed at 3s | Meta Decision Description |
|---|---|---|---|
| stationary | stationary | stationary | "Stay stationary." |
| stationary | moving | - | "Start moving soon." |
| stationary | stationary | moving | "Stay stationary for now, then start moving soon." |
| moving | constant | constant | "Keep speed." |
| moving | constant | increase | "Keep speed, then accelerate." |
| moving | constant | decrease | "Keep speed, then brake." |
| moving | increase | increase | "Accelerate." |
| moving | increase | constant | "Accelerate, then keep high speed." |
| moving | increase | decrease | "Accelerate, then brake." |
| moving | decrease | decrease | "Brake." |
| moving | decrease | constant | "Brake, then keep low speed." |
| moving | decrease | increase | "Brake, then accelerate." |

Table 6: Summary of 12 categories of the meta decision in chain-of-thought reasoning. We use speed at 3 different future timestamps, *i.e.*, at future 0, 1, and 3 seconds, as the decision points. If we are able to identify the cause of speed changes, *e.g.*, due to traffic signs or critical objects, then we also append it to the description.

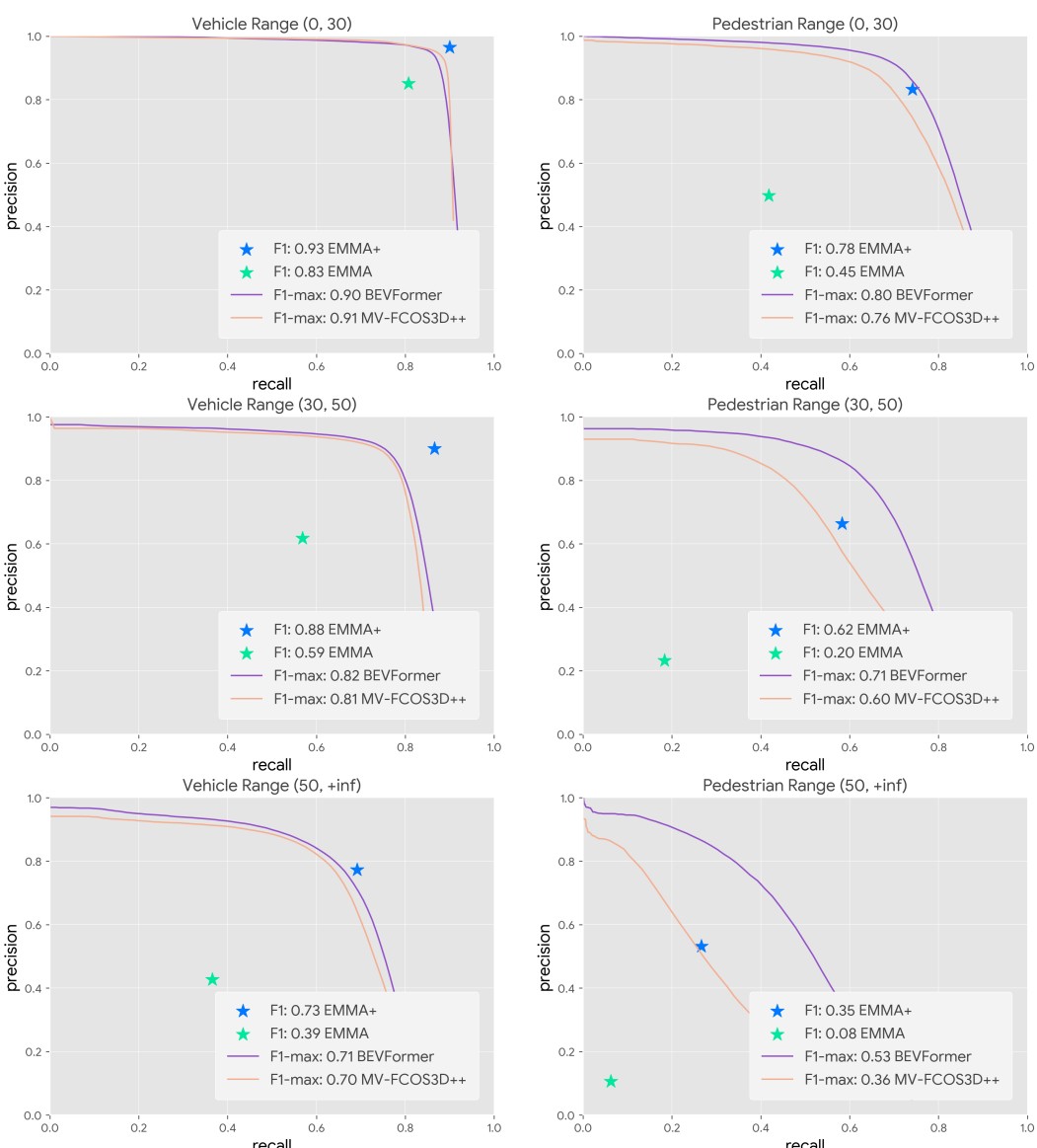

Figure 11: Distance breakdowns of the camera-primary 3D object detection experiments on WOD (Sun et al., 2020) using the standard LET matching (Hung et al., 2024). We observe that the performance gap between EMMA+ and the baseline models diminishes as the distance to objects increases. We attribute this phenomenon to the potentially lower resolution of the camera input used by EMMA compared to that of the baseline models.

## A.3    Failure Examples

While Section 3.6 showcases numerous successful predictions from EMMA across trajectory, detection, and road graph tasks, it is equally important to analyze its limitations. To provide insights for future development, we present three distinct failure scenarios in Figure 12, each illustrating a different type of error.

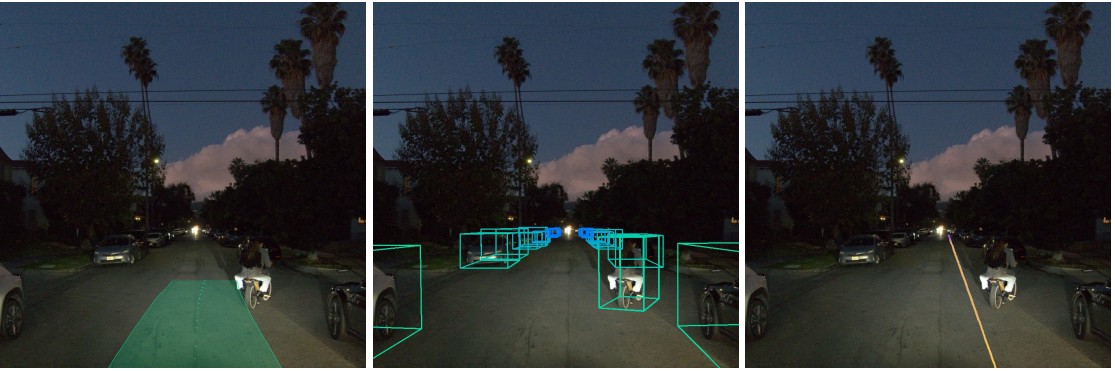

EMMA's predicted trajectory exhibits a suboptimal clearance from the motorbike. While the prediction is validated when the motorbike later tracks further right, the ideal path would have included a greater initial lateral offset.

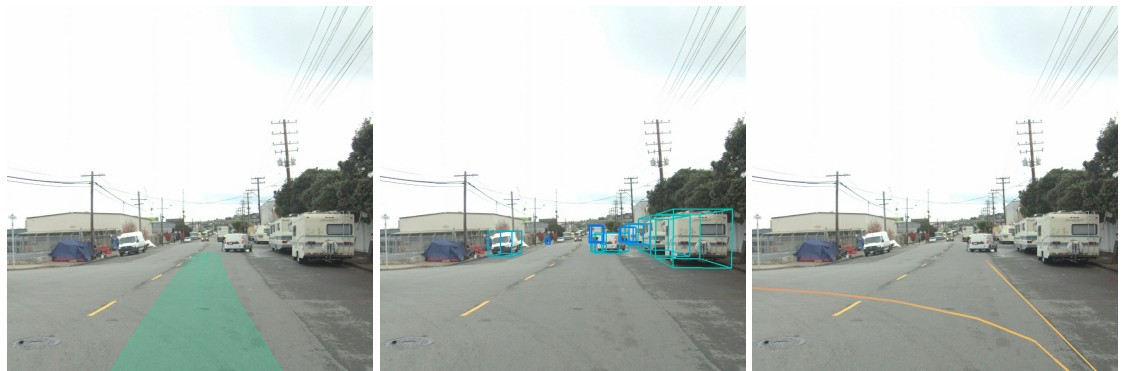

EMMA fails to detect a distant oncoming vehicle, though it is correctly identified in the subsequent frame. This one-frame detection delay is suboptimal, as the ego vehicle plans to nudging to the left. Earlier detection would have led to a safer and more controlled trajectory.

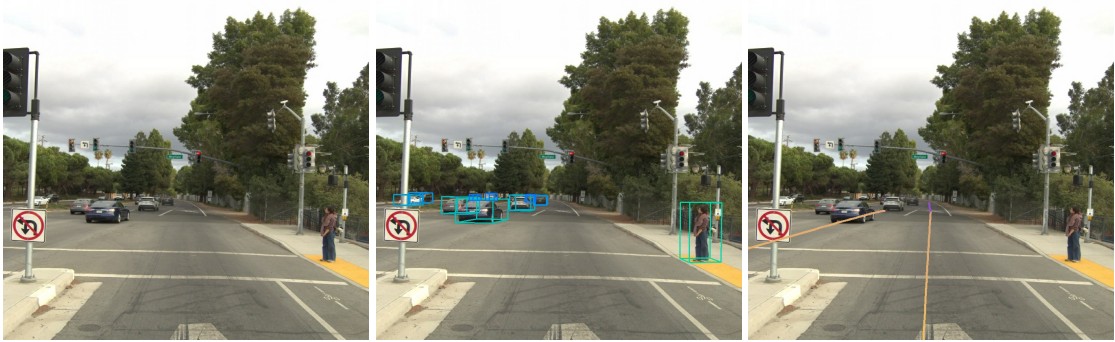

While EMMA correctly identifies the immediate drivable lanes, it fails to detect an upcoming lane merge further down the road. This oversight may be attributed to the degraded painted arrow.

Figure 12: Failure examples of EMMA prediction. Each row contains a scenario where EMMA's predictions were partially flawed: end-to-end planning trajectory (left), 3D object detection (middle), and road graph estimation (right).

## A.4 Concrete Prompts and Answers

As demonstrated in Section 2.3 and Figure 2, EMMA is capable of generating desired outputs based on varying prompts. To facilitate reproducibility, Table 7 provides the specific prompts used and their corresponding model-predicted answers.

| Image | Prompt and Answer |
|---|---|
| 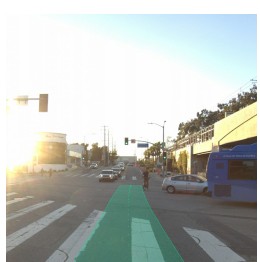 | **Prompt:** Assume I am driving at the coordinate 0.0, 0.0. The high-level behavior attention is: go straight. The past trajectory under vehicle coordinate is: -4.09, 0.01 and -3.94, 0.01 and -3.71, 0.01 and -3.42, 0.01 and -3.05, 0.00 and -2.60, 0.00 and -2.07, -0.00 and -1.45, -0.00 and -0.76, -0.00 and 0.00, 0.00. The past ego velocity under vehicle coordinate is: 0.94, -0.01 and 1.29, -0.01 and 1.66, -0.01 and 2.04, -0.02 and 2.45, -0.02 and 2.88, -0.02 and 3.26, -0.02 and 3.65, -0.01 and 3.98, -0.00. The past ego acceleration under vehicle coordinate is: 0.32, -0.00 and 0.34, -0.00 and 0.37, -0.00 and 0.38, -0.00 and 0.40, -0.00 and 0.43, 0.00 and 0.38, 0.00 and 0.39, 0.01 and 0.34, 0.01. What is my future trajectory in next 5 seconds under vehicle coordinate? 

 **Answer:** 0.83, 0.01 and 1.72, 0.01 and 2.67, 0.02 and 3.67, 0.04 and 4.73, 0.05 and 5.84, 0.07 and 6.95, 0.09 and 8.08, 0.11 and 9.23, 0.13 and 10.41, 0.15 and 11.62, 0.17 and 12.87, 0.19 and 14.15, 0.22 and 15.47, 0.24 and 16.80, 0.26 and 18.15, 0.27 and 19.52, 0.28 and 20.92, 0.28 and 22.36, 0.28 and 23.83, 0.28 and 25.34, 0.27 and 26.89, 0.26 and 28.49, 0.25 and 30.14, 0.23 and 31.83, 0.22 |
| 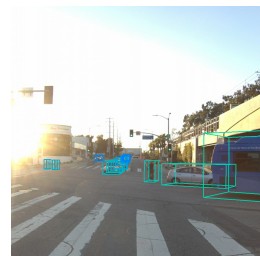 | **Prompt:** What are the 3D bounding boxes for objects in the following list: vehicle, pedestrian, and cyclist? Describe the bounding boxes in the format of [center_x, center_y, bottom_z, width, length, height, heading, class], where heading is in radian, class in text, and other dimensions are in meter? 

 **Answer:** -12.91 -9.23 -0.21 12.99 3.21 3.45 -2.25 vehicle and 21.81 -4.15 -0.15 4.61 2.10 1.65 -2.18 vehicle and 25.61 -1.11 -0.12 1.89 0.86 1.86 -0.23 cyclist and 39.12 3.25 -0.13 4.39 2.13 1.79 3.14 vehicle and 45.51 3.18 -0.12 4.65 2.10 1.54 3.14 vehicle and 43.81 21.51 -0.01 4.71 2.19 1.51 -2.16 vehicle and 40.01 20.11 0.09 4.92 2.39 2.02 -2.17 vehicle and 44.81 3.16 -0.11 4.51 2.10 1.55 3.14 vehicle and 49.11 13.39 0.02 1.11 0.83 1.68 -0.05 pedestrian and 52.31 3.02 0.05 4.85 2.23 1.92 -3.14 vehicle and 51.11 15.41 0.05 1.12 0.85 1.78 -0.01 pedestrian and 62.51 3.35 0.49 4.51 2.15 1.85 -3.14 vehicle and 72.11 2.99 0.92 4.45 2.13 1.74 -3.13 vehicle and 79.21 3.41 1.19 4.41 2.14 1.72 -3.13 vehicle and 85.51 3.35 1.45 4.36 2.08 1.60 -3.13 vehicle and 85.11 11.11 1.41 4.49 2.15 1.76 3.11 vehicle and 92.21 3.51 1.69 4.45 2.18 1.90 -3.13 vehicle |
| 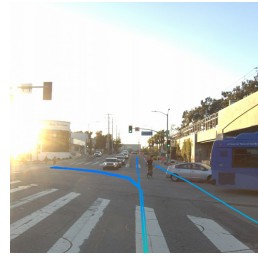 | **Prompt:** My current position is 1.28, -0.00. Where are driveable lanes in front of me? Describe the lanes as a series of points relative to 0.00, 0.00? 

 **Answer:** (0.00, -0.11 and 10.00, -0.11 and 20.00, -0.11 and 30.00, -0.11 and 40.00, -0.11 and 51.11, -0.11 and invalid and invalid and invalid and invalid) valid ; (51.11, -0.11 and 61.10, -0.06 and 71.09, -0.01 and 81.08, 0.04 and 91.07, 0.09 and invalid and invalid and invalid and invalid and invalid) valid ; (0.00, -0.12 and 10.00, -0.19 and 15.11, -0.22 and 21.99, 0.62 and 27.72, 2.16 and 33.11, 4.65 and 38.02, 8.00 and 42.35, 12.05 and invalid and invalid) valid ; (0.00, -0.12 and 10.00, -0.13 and 17.11, -0.14 and 23.01, 0.58 and 28.76, 2.09 and 34.16, 4.56 and 39.07, 7.91 and 43.43, 11.95 and invalid and invalid) valid ; (0.00, -0.12 and 10.00, -0.13 and 16.11, -0.14 and 22.01, 0.55 and 27.76, 2.05 and 33.14, 4.55 and 38.00, 7.98 and 42.29, 12.09 and invalid and invalid) valid ; (0.00, -3.31 and 10.00, -3.25 and 20.00, -3.19 and 30.00, -3.13 and 40.00, -3.07 and 49.35, -3.01 and invalid and invalid and invalid and invalid) valid ; (invalid and invalid and invalid and invalid and invalid and invalid and invalid and invalid and invalid and invalid) invalid ... |

Table 7: The specific prompts used and their corresponding model-predicted answers. Numerical values are color-coded in blue and predicted separators in red for better visualization.

### A.5 Limitations, Risks, and Mitigations

In the main paper, we demonstrate state-of-the-art end-to-end motion planning on the nuScenes planning benchmark. We also achieve competitive performance for camera-primary 3D detection on WOD. Furthermore, our generalist setup improves the quality across multiple tasks through joint training. Despite these promising results, we acknowledge the limitations of our work and propose directions for building on this foundation and addressing such challenges in future research.

**Memory and video capability**: Currently, our model processes only a limited number of frames (up to 4), restricting its ability to capture the long-term dependencies essential for driving tasks. Effective driving requires not just real-time decision-making but also reasoning over extended time horizons, relying on long-term memory to anticipate and respond to evolving scenarios. Enhancing the model's ability to perform long-term reasoning is a promising area for future research. This could potentially be achieved by integrating memory modules or extending its capability to process longer video sequences efficiently, enabling more comprehensive temporal understanding.

**Extension to LiDAR and radar input**: Our approach heavily relies on pre-trained MLLMs, which typically do not incorporate LiDAR or Radar inputs. Expanding our model to integrate these 3D sensing modalities presents two key challenges: **1)** There is a significant imbalance between the volume of available camera and 3D sensing data, resulting in less generalizable 3D sensing encoders as compared to their camera-based counterparts. **2)** The development of 3D sensing encoders has not yet reached the scale and sophistication of camera-based encoders. A potential solution to address these challenges is to pre-train a large-scale 3D sensing encoder using data carefully aligned with camera inputs. This approach may foster better cross-modality synergy and substantially improve the generalization capabilities of the 3D sensing encoder.

**Verification of the predicted driving signals**: Our model can directly predict driving signals without relying on intermediate outputs, such as object detection or road graph estimation. This approach introduces challenges for both real-time and post-hoc verification. We have demonstrated that our generalist model can jointly predict additional human readable outputs such as objects and road graph elements, and the driving decision can be further explained with chain-of-thought driving rationale. However, there is no guarantee that these outputs will be always consistent despite the empirical observations that they are often indeed consistent. Improving driving rationale is one of our future research directions.

**Sensor simulation for closed-loop evaluation**: To accurately assess an end-to-end autonomous driving system in a closed-loop environment, a comprehensive sensor simulation solution is necessary. However, the computational cost of sensor simulation is often much higher than that of behavior simulators. This significant cost burden can hinder thorough testing and verification of an end-to-end models; however, the field of efficient sensor simulation methods continues to rapidly evolve, with the promise of significantly mitigating this burden.

**More targeted and accurate open-loop evaluation**: While existing open-loop evaluation offers low computational cost, its results are sometimes unreliable. The popular nuScenes (Caesar et al., 2020) benchmark, for instance, exhibits well-documented limitations. Key metrics like collision rate are sensitive to hyperparameter choices, such as the BEV grid resolution (Weng et al., 2024; Zhai et al., 2023). Furthermore, many of its scenarios lack planning diversity and can be trivially solved by extrapolating historical trajectories. Initiatives like NAVSIM (Dauner et al., 2024) are beginning to address these vulnerabilities, but this highlights a critical need for future work on developing more challenging and trustworthy open-loop evaluation frameworks.

**Challenges of onboard deployment**: Autonomous driving demands real-time decision-making, which poses a significant challenge when deploying large models due to their increased inference latency. This creates a need for optimizing the model or distilling it into a more compact form suitable for deployment, all while maintaining performance and safety standards.

While the main paper is focused on establishing the EMMA architecture, the framework is amenable to various well-established optimization techniques. To illustrate this adaptability, we have experimented with a latency-optimized EMMA configuration. By incorporating strategies such as SARA-RT (Leal et al., 2024),

employing shorter action sequences, streamlining by removing explicit reasoning chains, *etc.*, this variant achieves an inference speed of 3 FPS, a 67% speedup compared to UniAD's 1.8 FPS. This preliminary result underscores that the core EMMA architecture possesses the flexibility to be effectively adapted into specialized, low-latency variants. Achieving a delicate balance between model size, efficiency, and quality is crucial for the successful real-world deployment of autonomous driving systems, and represents a key area for future research.

