# OpenReview forum: "EMMA: End-to-End Multimodal Model for Autonomous Driving"
_TMLR — Accepted by TMLR_

### Review · Reviewer_C53W · 2025-04-03

**Summary Of Contributions:**

The authors built an e2e autonomous driving model using multimodal large language models.
The proposed model takes video data and high-level commands as input and generates trajectories of future waypoints.
The model can be trained in a self-supervised manner (only ego-vehicle trajectories as targets) or as a generalist model that also performs auxiliary perception tasks and generates a text-based driving rationale to improve its interpretability.

**Audience:**

Yes

**Claims And Evidence:**

Yes

**Requested Changes:**

See weaknesses.

**Strengths And Weaknesses:**

Strengths:
- Generating a driving rationale improves the interpretability of e2e planning models beyond auxiliary perception outputs.
- The proposed hierarchical structure of the driving rationale is intuitive and well adapted to the different levels of behavior generation in autonomous driving.
- The authors perform experiments on nuScenes, Waymo Open Motion, Waymo Open Dataset and a very large internal dataset. The results suggest that their generalist model is able to outperform specialized models on e2e motion planning and object detection.
- Notably, the simplest self-supervised version of EMMA (cf. section 3.1) is able to outperform related methods that require more supervision during training.
- They ablate their design choices, leading to interesting insights (e.g., that adding punctuation to sequences improves results).

Weaknesses:
- The description of how the text-based driving rationale is generated lacks details. For example, what are the carefully designed visual and text prompts, or what heuristic algorithm is used to generate meta driving decisions?
- The road graph estimation (section 3.3.1) is not compared to other methods. A comparison with related work on map perception (Liao et al., 2023) or the map perception outputs of e2e models like UniAD would strengthen the paper.
- The main results in table 1 highlight the model versions, which are trained in a self-supervised manner. Therefore, the related work section should include a paragraph on recent self-supervised learning methods for autonomous driving (e.g., for scene reconstruction (Yang et al., 2024), motion forecasting (Wagner et al., 2024), and world models (Zhang et al., 2024)).

Liao et al., "MapTR: Structured Modeling and Learning for Online Vectorized HD Map Construction" ICLR 2023

Yang et al. "EmerNeRF: Emergent Spatial-Temporal Scene Decomposition via Self-Supervision" ICLR 2024

Wagner et al. "JointMotion: Joint Self-Supervision for Joint Motion Prediction" CoRL 2024

Zhang et al. "Copilot4D: Learning Unsupervised World Models for Autonomous Driving via Discrete Diffusion" ICLR 2024

---

### Review · Reviewer_3oH1 · 2025-04-06

**Summary Of Contributions:**

The EMMA paper introduces a novel end-to-end autonomous driving model that unifies motion planning, 3D object detection, and road graph estimation within a single large multimodal language model framework. By representing both inputs (multi-camera images) and outputs (trajectories, objects, maps) in natural language, EMMA leverages the reasoning and generalization capabilities of a pre-trained foundation model (Gemini) without relying on LiDAR, HD maps, or intermediate perception modules. It further incorporates chain-of-thought (CoT) reasoning to explicitly describe the decision-making process behind trajectory predictions, leading to measurable improvements in planning accuracy. Through multitask co-training and large-scale vision-language pretraining, EMMA achieves state-of-the-art performance on nuScenes and Waymo benchmarks using only camera data, while offering enhanced interpretability and flexibility across tasks.

**Audience:**

Yes

**Claims And Evidence:**

Yes

**Requested Changes:**

1. Authors should include **collision rate** metrics in Table 1. Compared to L2 errors, collision statistics provide a more intuitive and safety-relevant measure of planning quality.

2. The paper would benefit from including **failure cases** to illustrate the model’s current limitations. While several successful qualitative examples are shown, presenting scenarios where EMMA fails—such as inaccurate trajectory predictions, misaligned reasoning outputs, or poor handling of rare cases—would provide a more balanced evaluation and help clarify the boundaries of the model’s generalization capabilities.

3. The paper should include a discussion on potential **latency** issues associated with LLM-based planners, as planning speed is a critical constraint in real-time autonomous driving systems. Given EMMA is built on a large multimodal language model, inference latency may limit its deployment in high-speed or latency-sensitive scenarios.

4. The paper would be improved by including **concrete prompt and output examples** for each of the tasks, such as planning, object detection, and road graph estimation.

5. By the way,  do the authors plan to make public the internal dataset you use in the future? I think this is far more significant than training a model

**Strengths And Weaknesses:**

**Strengths:**

EMMA stands out for its unified multimodal architecture, combining motion planning, object detection, and road graph estimation into a single vision-language model, which eliminates the need for separate perception modules and enhances generalization across various tasks. Another strength is its incorporation of chain-of-thought reasoning, where the model generates structured natural language explanations for its decisions, leading to a significant 6.7% improvement in planning accuracy. Finally, EMMA demonstrates state-of-the-art performance using only monocular camera input, outperforming many models that rely on LiDAR or HD maps, showcasing the effectiveness of pretraining large multimodal models for autonomous driving tasks.

**Weakness**:

1. Can authors provide more concrete details about the internal dataset used for pretraining and evaluation? The paper does not specify the scale, the types of long-tail scenarios included, or how diverse driving behaviors are captured. Since EMMA+ benefits significantly from this internal data, understanding how it was constructed—especially regarding navigation prompt design and scenario diversity—is crucial for evaluating the model’s planning results on Nuscenes dataset.

2. Can authors clarify the training configurations used for each experimental result in Section 3? While Section 2 clearly outlines the objective functions for different tasks, it remains unclear how the results in different tables were obtained. For example, in Table 1, the planning results appear to come from training EMMA only on planning data, whereas in Section 3.3, EMMA+ is pre-trained on the 3D detection task. Although the paper emphasizes EMMA as a unified model, many experiments seem to be conducted with task-specific training, which makes me confused.

3.  Can authors further clarify whether the end-to-end planning tasks in Section 3.4 include the reasoning component introduced earlier in the paper? I think that Section 3.4 is arguably the most important part of the experimental analysis, as it directly evaluates the effect of each task on the performance of the unified driving model. While the paper provides a clear breakdown of how different tasks contribute to planning, detection, and mapping, it remains unclear whether chain-of-thought reasoning is integrated during multitask training. One of the key advantages of using a vision-language model for autonomous driving lies in its explainability and open-vocabulary reasoning capabilities. However, these strengths are not sufficiently discussed or evaluated. Although Table 2 shows that reasoning yields some improvements in planning metrics, this gain is measured in an open-loop setting, where the practical value of such improvement is limited.

---

### Review · Reviewer_HGfn · 2025-04-28

**Summary Of Contributions:**

EMMA introduces a unified, end-to-end multimodal model for autonomous driving built on a large multimodal language model foundation, which directly maps raw camera data and text inputs to key driving outputs such as planner trajectories, 3D object detections, and road graph elements. By representing all non-sensor inputs and outputs as natural language, EMMA enables joint processing of diverse driving tasks in a single language space, supporting flexible prompting and co-training across tasks. The model achieves cutting-edge results in end-to-end motion planning on public benchmarks, as well as competitive performance in 3D object detection and road graph estimation. EMMA’s generalist design allows a single model to match or surpass the performance of individually trained models on multiple autonomous driving tasks, demonstrating strong scalability and robustness. The integration of a CoT reasoning module further enhances both planning performance and model explainability.

**Audience:**

Yes

**Broader Impact Concerns:**

* The model’s ability to generate detailed scene descriptions (via chain-of-thought rationales) raises privacy risks if deployed in real-time systems, particularly when combined with surveillance-capable sensors.
* Accountability Gaps. As a generalist model producing multiple outputs (trajectories, object detections, road graphs), EMMA complicates liability attribution in accidents. Existing legal frameworks struggle to assign responsibility between developers, manufacturers, and users when AI systems fail.
* Widespread adoption of EMMA-based systems could disrupt transportation-sector jobs, disproportionately affecting drivers in logistics and ride-hailing services.

**Claims And Evidence:**

Yes

**Requested Changes:**

* Open-source the EMMA model, its dataset, or its benchmark to enable broader access for the research community, facilitate further research and development, and enhance the overall impact of the work.
* Extend EMMA’s temporal reasoning capabilities by enabling longer video sequence processing, larger context window, or integrating memory modules.
* Investigate methods to reduce computational overhead for real-time deployment.
* Develop formal mechanisms to verify consistency across predicted outputs and enhance safety assurances.

**Strengths And Weaknesses:**

## Strengths
* This paper presents a thorough set of experiments across multiple autonomous driving tasks and provides clear, well-structured comparisons with existing methods. The results are systematically reported, and the manuscript is written in a clear and accessible manner, making it easy to follow the design and understand the full picture.
* It presents a unified end-to-end model that reframes multiple autonomous driving tasks-including motion planning, 3D object detection, and road graph estimation-as vision-language problems using a MLLM foundation, enabling joint processing and co-training across tasks for improved efficiency and performance.
* The model leverages the broad world knowledge and generalization capabilities of MLLMs, allowing it to better handle rare and complex driving scenarios that are underrepresented in standard autonomous driving datasets; as training data and foundation models scale, EMMA’s performance and robustness continue to improve, especially in long-tail situations.
* It achieves SOTA results in end-to-end motion planning on public benchmarks such as nuScenes, and demonstrates competitive or superior performance in 3D object detection and road graph estimation compared to existing methods.
* The integration of a hierarchical CoT reasoning module requires EMMA to generate structured rationales-including scene descriptions, identification and behavior of critical objects, and meta driving decisions-prior to trajectory prediction, which not only boosts planning quality but also enhances the explainability of the model’s decisions.
 ---

## Weaknesses
* The impact of this work could be significantly increased by open-sourcing the model, dataset, or benchmark, enabling broader access and fostering further research and development within the community.
* Limited number of image frames restricts its ability to capture long-term dependencies and hinders reasoning about evolving or complex scenarios that require extended temporal context.
* Although EMMA can jointly predict driving signals and human-readable outputs such as object lists and road graphs, there is no formal guarantee of outputs consistency, posing challenges for both real-time and post-hoc verification of the model’s decision-making-a critical issue for autonomous driving safety.

---

### Review · Reviewer_DKk7 · 2025-05-04

**Summary Of Contributions:**

The paper introduces EMMA, an end-to-end multimodal model for autonomous driving built on top of Gemini. Instead of using modular designs, it directly maps raw observations to desired scene understanding and planning outputs in the form of natural language. Through a hierarchical reasoning process, the resulting model achieves state-of-the-art planning results on both nuScenes and an in-house large-scale benchmark. The paper also explores the benefits of co-training with other perception tasks, showing the potential of this unified framework.

**Audience:**

Yes

**Broader Impact Concerns:**

None.

**Claims And Evidence:**

Yes

**Requested Changes:**

C1) It would be beneficial to report and compare the inference speed.

C2) Evaluating planning performance on nuScenes is known to be unstable [1,2]. Testing on more recent benchmarks like NAVSIM would be more convincing.

---

[1] Rethinking the Open-Loop Evaluation of End-to-End Autonomous Driving in nuScenes

[2] Is Ego Status All You Need for Open-Loop End-to-End Autonomous Driving?

**Strengths And Weaknesses:**

**Strengths**

S1) The authors make a pioneering attempt to unify all driving tasks as a language processing formulation. The proposed framework is simple yet powerful, achieving state-of-the-art performance.

S2) Sufficient details are provided to help the audience understand the behavior of the model.

S3) The authors also demonstrate the benefits of chain-of-thought reasoning and co-training, which will inspire future research in developing a generalist driving agent.

S4) The presentation is clear and easy to follow.

**Weaknesses**

W1) My biggest concern is the latency of this framework. The paper does not compare previous methods in terms of inference speed. Autoregressively predicting chain-of-thought may exacerbate this issue, which contradicts real-world applications.

W2) The proposed method is developed based on a private Gemini model using in-house datasets, making the community hard to follow.

W3) From my understanding, the chain-of-thought component requires labels from other models. Could you clarify why EMMA can still be considered a self-supervised method in this case?

---

### Decision · Action_Editor_KChu · 2025-06-18

**Recommendation:** Accept as is

**Audience:**

Yes

**Audience Explanation:**

This paper tackles the important problems in autonomous driving, which itself is of interest. The MLLM approach is a nature fit this problem. The work is well executed, with clear experiments and systematic comparisons to existing methods.

The empirical findings are insightful: (1) model generalization in rare and complex driving scenarios, (2) CoT improves planning quality and explainability

**Claims And Evidence:**

Yes

**Claims Explanation:**

This paper introduces EMMA, a unified end-to-end Multimodal Large Language Model (MLLM) for autonomous driving that formulates tasks like motion planning, 3D object detection, and road graph estimation as vision-language problems. Key findings include:

- Joint training and co-processing across tasks improve both efficiency and performance.
- EMMA leverages the generalization strengths of MLLMs, excelling in rare and complex driving scenarios, especially as model and data scale.
- It achieves SOTA results on benchmarks like nuScenes and performs competitively in 3D detection and road graph tasks.
- A hierarchical CoT reasoning module enables EMMA to generate structured rationales (e.g., scene descriptions, object behaviors, and decisions) before predicting trajectories, which enhances planning quality and explainability.